# PARALLEL DEEP NEURAL NETWORKS HAVE ZERO DUALITY GAP

**Yifei Wang, Tolga Ergen & Mert Pilanci**
Department of Electrical Engineering
Stanford University
{wangyf18,ergen,pilanci}@stanford.edu

## ABSTRACT

Training deep neural networks is a challenging non-convex optimization problem. Recent work has proven that the strong duality holds (which means zero duality gap) for regularized finite-width two-layer ReLU networks and consequently provided an equivalent convex training problem. However, extending this result to deeper networks remains to be an open problem. In this paper, we prove that the duality gap for deeper linear networks with vector outputs is non-zero. In contrast, we show that the zero duality gap can be obtained by stacking standard deep networks in parallel, which we call a parallel architecture, and modifying the regularization. Therefore, we prove the strong duality and existence of equivalent convex problems that enable globally optimal training of deep networks. As a by-product of our analysis, we demonstrate that the weight decay regularization on the network parameters explicitly encourages low-rank solutions via closed-form expressions. In addition, we show that strong duality holds for three-layer standard ReLU networks given rank-1 data matrices.

## 1 INTRODUCTION

Deep neural networks demonstrate outstanding representation and generalization abilities in popular learning problems ranging from computer vision, natural language processing to recommendation system. Although the training problem of deep neural networks is a highly non-convex optimization problem, simple first order gradient based algorithms, such as stochastic gradient descent, can find a solution with good generalization properties. However, due to the non-convex and non-linear nature of the training problem, underlying theoretical reasons for this remains an open problem.

The Lagrangian dual problem (Boyd et al., 2004) plays an important role in the theory of convex and non-convex optimization. For convex optimization problems, the convex duality is an important tool to determine their optimal values and to characterize the optimal solutions. Even for a non-convex primal problem, the dual problem is a convex optimization problem the can be solved efficiently. As a result of weak duality, the optimal value of the dual problem serves as a non-trivial lower bound for the optimal primal objective value. Although the duality gap is non-zero for non-convex problems, the dual problem provides a convex relaxation of the non-convex primal problem. For example, the semi-definite programming relaxation of the two-way partitioning problem can be derived from its dual problem (Boyd et al., 2004).

The convex duality also has important applications in machine learning. In Paternain et al. (2019), the design problem of an all-encompassing reward can be formulated as a constrained reinforcement learning problem, which is shown to have zero duality. This property gives a theoretical convergence guarantee of the primal-dual algorithm for solving this problem. Meanwhile, the minimax generative adversarial net (GAN) training problem can be tackled using duality (Farnia & Tse, 2018).

In lines of recent works, the convex duality can also be applied for analyzing the optimal layer weights of two-layer neural networks with linear or ReLU activations (Ergen & Pilanci, 2019; Pilanci & Ergen, 2020; Ergen & Pilanci, 2020a;b; Lacotte & Pilanci, 2020; Sahiner et al., 2020). Based on the convex duality framework, the training problem of two-layer neural networks with ReLU activation can be represented in terms of a single convex program in Pilanci & Ergen (2020). Such convex optimization formulations are extended to two-layer and three-layer convolutional neural

network training problems in Ergen & Pilanci (2021b). Strong duality also holds for deep linear neural networks with scalar output (Ergen & Pilanci, 2021a). The convex optimization formulation essentially gives a detailed characterization of the global optimum of the training problem. This enables us to examine in numerical experiments whether popular optimizers for neural networks, such as gradient descent or stochastic gradient descent, converge to the global optimum of the training loss.

Admittedly, a zero duality gap is hard to achieve for deep neural networks, especially for those with vector outputs. This imposes more difficulty to understand deep neural networks from the convex optimization lens. Fortunately, neural networks with parallel structures (also known as multi-branch architecture) appear to be easier to train. Practically, the usage of parallel neural networks dates back to AlexNet (Krizhevsky et al., 2012). Modern neural network architecture including Inception (Szegedy et al., 2017), Xception (Chollet, 2017) and SqueezeNet (Iandola et al., 2016) utilize the parallel structure. As the "parallel" version of ResNet (He et al., 2016a;b), ResNeXt (Xie et al., 2017) and Wide ResNet (Zagoruyko & Komodakis, 2016) exhibit improved performance on many applications. Recently, it was shown that neural networks with parallel architectures have smaller duality gaps (Zhang et al., 2019) compared to standard neural networks. Furthermore, Ergen & Pilanci (2021c;e) proved that there is no duality gap for parallel architectures with three-layers.

On the other hand, it is known that overparameterized parallel neural networks have benign training landscapes (Haeffele & Vidal, 2017; Ergen & Pilanci, 2019). The parallel models with the over-parameterization are essentially neural networks in the mean-field regime (Nitanda & Suzuki, 2017; Mei et al., 2018; Chizat & Bach, 2018; Mei et al., 2019; Rotskoff et al., 2019; Sirignano & Spiliopoulos, 2020; Akiyama & Suzuki, 2021; Chizat, 2021; Nitanda et al., 2020). The deep linear model is also of great interests in the machine learning community. For training $\ell_2$ loss with deep linear networks using Schatten norm regularization, Zhang et al. (2019) show that there is no duality gap. The implicit regularization in training deep linear networks has been studied in Ji & Telgarsky (2018); Arora et al. (2019); Moroshko et al. (2020). From another perspective, the standard two-layer network is equivalent to the parallel two-layer network. This may also explain why there is no duality gap for two-layer neural networks.

## 1.1 CONTRIBUTIONS

Following the convex duality framework introduced in Ergen & Pilanci (2021a; 2020a), which showed the duality gap is zero for two-layer networks, we go beyond two-layer and study the convex duality for vector-output deep neural networks with linear activation and ReLU activation. **Surprisingly, we prove that three-layer networks may have duality gaps depending on their architecture, unlike two-layer neural networks which always have zero duality gap**. We summarize our contributions as follows.

- For training standard vector-output deep linear networks using $\ell_2$ regularization, we precisely calculate the optimal value of the primal and dual problems and show that the **duality gap is non-zero**, i.e., Lagrangian relaxation is inexact. We also demonstrate that the $\ell_2$-regularization on the parameter explicitly forces a tendency toward a low-rank solution, which is boosted with the depth. However, we show that the optimal solution is available in **closed-form**.

- For parallel deep linear networks, with certain convex regularization, we show that **the duality gap is zero**, i.e, Lagrangian relaxation is exact.

- For parallel deep ReLU networks of arbitrary depth, with certain convex regularization and sufficient number of branches, we prove strong duality, i.e., show that **the duality gap is zero**. Remarkably, this guarantees that **there is a convex program equivalent to the original deep ReLU neural network problem.**

We summarize the duality gaps for parallel/standard neural network in Table 1.

## 1.2 NOTATIONS

We use bold capital letters to represent matrices and bold lowercase letters to represent vectors. Denote $[n] = \{1, \ldots, n\}$. For a matrix $\mathbf{W}_l \in \mathbb{R}^{m_{l-1} \times m_l}$, for $i \in [m_{l-1}]$ and $j \in [m_l]$, we denote $\mathbf{w}_{l,i}^{\text{col}}$ as its

|  | linear activation | | | ReLU activation | | |
|---|---|---|---|---|---|---|
|  | $L = 2$ | $L = 3$ | $L > 3$ | $L = 2$ | $L = 3$ | $L > 3$ |
| **standard networks** | | | | | | |
| previous work | ✓(0) | ✗ | ✗ | ✓(0) | ✗ | ✗ |
| this paper | ✓(0) | ✓($\neq 0$) | ✓($\neq 0$) | ✓(0) | ✗ | ✗ |
| **parallel networks** | | | | | | |
| previous work | ✓(0) | ✓(0) | ✓(0) | ✓(0) | ✓(0) | ✗ |
| this paper | ✓(0) | ✓(0) | ✓(0) | ✓(0) | ✓(0) | ✓(0) |

Table 1: Existing and current results for duality gaps in $L$-layer standard and parallel architectures. we compare our duality gap characterization with previous literature. Each check mark indicates whether a characterization of the duality gap exists for the corresponding architecture and the number next to it indicates whether the gap is zero or not.

$i$-th column and $\mathbf{w}_{l,j}^{\text{row}}$ as its $j$-th row. Throughout the paper, $\mathbf{X} \in \mathbb{R}^{N \times d}$ is the data matrix consisting of $d$ dimensional $N$ samples and $\mathbf{Y} \in \mathbb{R}^{N \times K}$ is the label matrix for a regression/classification task with $K$ outputs. We use the letter $P$ $(D)$ for the optimal value of the primal (dual) problem.

## 1.3 MOTIVATIONS AND BACKGROUND

Recently a series of papers (Pilanci & Ergen, 2020; Ergen & Pilanci, 2021a; 2020a) studied two-layer neural networks via convex duality and proved that strong duality holds for these architectures. Particularly, these prior works consider the following weight decay regularized training framework for classification/regression tasks. Given a data matrix $\mathbf{X} \in \mathbb{R}^{N \times d}$ consisting of $d$ dimensional $N$ samples and the corresponding label matrix $\mathbf{y} \in \mathbb{R}^N$, the weight-decay regularized training problem for a scalar-output neural network with $m$ hidden neurons can be written as follows

$$P := \min_{\mathbf{W}_1, \mathbf{w}_2} \frac{1}{2} \|\phi(\mathbf{X}\mathbf{W}_1)\mathbf{w}_2 - \mathbf{y}\|_2^2 + \frac{\beta}{2}(\|\mathbf{W}_1\|_F^2 + \|\mathbf{w}_2\|_2^2), \tag{1}$$

where $\mathbf{W}_1 \in \mathbb{R}^{d \times m}$ and $\mathbf{w}_2 \in \mathbb{R}^m$ are the layer weights, $\beta > 0$ is a regularization parameter, and $\phi$ is the activation function, which can be linear $\phi(z) = z$ or ReLU $\phi(z) = \max\{z, 0\}$. Then, one can take the dual of (1) with respect to $\mathbf{W}_1$ and $\mathbf{w}_2$ obtain the following dual optimization problem

$$D := \max_{\boldsymbol{\lambda}} \ -\frac{1}{2}\|\boldsymbol{\lambda} - \mathbf{y}\|_2^2 + \frac{1}{2}\|\mathbf{y}\|_2^2,$$
$$\text{s.t.} \ \max_{\mathbf{w}_1 : \|\mathbf{w}_1\|_2 \leq 1} |\boldsymbol{\lambda}^T \phi(\mathbf{X}\mathbf{w}_1)| \leq \beta. \tag{2}$$

We first note that since the training problem (1) is non-convex, strong duality may not hold, i.e., $P \geq D$. Surprisingly, as shown in Pilanci & Ergen (2020); Ergen & Pilanci (2021a; 2020a), strong duality in fact holds, i.e., $P = D$, for two-layer networks and therefore one can derive exact convex representations for the non-convex training problem in (1). However, extensions of this approach to deeper and state-of-the-art architectures are not available in the literature. Based on this observation, the central question we address in this paper is:

*Does strong duality hold for deep neural networks?*

Depending on the answer to the question above, an immediate next questions we address is

*Can we characterize the duality gap (P-D)? Is there an architecture for which strong duality holds regardless of the depth?*

Consequently, throughout the paper, we provide a full characterization of convex duality for deeper neural networks. We observe that the dual of the convex dual problem of the nonconvex minimum norm problem of deep networks correspond to a minimum norm problem of deep networks with parallel branches. Based on this characterization, we propose a modified architecture for which strong duality holds regardless of depth.

### 1.4 ORGANIZATION

This paper is organized as follows. In Section 2, we review standard neural networks and introduce parallel architectures. For deep linear networks, we derive primal and dual problems for both standard and parallel architectures and provide calculations of optimal values of these problems in Section 3. We derive primal and dual problems for three-layer ReLU networks with standard architecture and precisely calculate the optimal values for whitened data in Section 4. We also show that deep ReLU networks with parallel structures have no duality gap.

## 2 STANDARD NEURAL NETWORKS VS PARALLEL ARCHITECTURES

We briefly review the convex duality theory for two-layer neural networks in Appendix A. To extend the theory to deep neural networks, we fist consider the $L$-layer neural network with the standard architecture:

$$f_{\boldsymbol{\theta}}(\mathbf{X}) = \mathbf{A}_{L-1}\mathbf{W}_L, \mathbf{A}_l = \phi(\mathbf{A}_{l-1}\mathbf{W}_l), \, \forall l \in [L-1], \mathbf{A}_0 = \mathbf{X}, \tag{3}$$

where $\phi$ is the activation function, $\mathbf{W}_l \in \mathbb{R}^{m_{l-1} \times m_l}$ is the weight matrix in the $l$-th layer and $\theta = (\mathbf{W}_1, \ldots, \mathbf{W}_L)$ represents the parameter of the neural network.

We then introduce the neural network with parallel architectures:

$$f_{\boldsymbol{\theta}}^{\mathrm{prl}}(\mathbf{X}) = \mathbf{A}_{L-1}\mathbf{W}_L, \mathbf{A}_{l,j} = \phi(\mathbf{A}_{l-1,j}\mathbf{W}_{l,j}), \forall l \in [L-1], \mathbf{A}_{0,j} = \mathbf{X}, \forall j \in [m]. \tag{4}$$

Here for $l \in [L-1]$, the $l$-th layer has $m$ weight matrices $\mathbf{W}_{l,j} \in \mathbb{R}^{m_{l-1} \times m_l}$ where $j \in [m]$. Specifically, we let $m_{L-1} = 1$ to make each parallel branch as a scalar-output neural network. In short, we can view the output $\mathbf{A}_{L-1}$ from a parallel neural network as a concatenation of $m$ scalar-output standard neural work. In Figures 1 and 2, we provide examples of neural networks with standard and parallel architectures. We shall emphasize that for $L = 2$, the standard neural network is identical to the parallel neural network. We next present a summary of our main result.

**Theorem 1 (main result)** *For $L \geq 3$, there exists an activation function $\phi$ and a $L$-layer standard neural network defined in (3) such that the strong duality does not hold, i.e., $P > D$. In contrast, for any $L$-layer parallel neural network defined in (4) with linear or ReLU activations and sufficiently large number of branches, strong duality holds, i.e., $P = D$.*

We elaborate on the primal problem with optimal value $P$ and the dual problem with optimal value $D$ in Section 3 and 4.

## 3 DEEP LINEAR NETWORKS

### 3.1 STANDARD DEEP LINEAR NETWORKS

We first consider the neural network with standard architecture, i.e., $f_{\boldsymbol{\theta}}(\mathbf{X}) = \mathbf{X}\mathbf{W}_1 \ldots \mathbf{W}_L$. Consider the following minimum norm optimization problem:

$$P_{\mathrm{lin}} = \min_{\{\mathbf{W}_l\}_{l=1}^L} \frac{1}{2} \sum_{l=1}^L \|\mathbf{W}_l\|_F^2, \tag{5}$$
$$\text{s.t. } \mathbf{X}\mathbf{W}_1, \ldots, \mathbf{W}_L = \mathbf{Y},$$

where the variables are $\mathbf{W}_1, \ldots, \mathbf{W}_L$. As shown in the Proposition 3.1 in (Ergen & Pilanci, 2021a), by introducing a scale parameter $t$, the problem (5) can be reformulated as

$$P_{\mathrm{lin}} = \min_{t>0} \frac{L-2}{2} t^2 + P_{\mathrm{lin}}(t),$$

where the subproblem $P_{\mathrm{lin}}(t)$ is defined as

$$P_{\mathrm{lin}}(t) = \min_{\{\mathbf{W}_l\}_{l=1}^L} \sum_{j=1}^K \|\mathbf{w}_{L,j}^{\mathrm{row}}\|_2,$$
$$\text{s.t. } \mathbf{X}\mathbf{W}_1 \ldots \mathbf{W}_L = \mathbf{Y}, \|\mathbf{W}_i\|_F \leq t, i \in [L-2], \|\mathbf{w}_{L-1,j}^{\mathrm{col}}\|_2 \leq 1, j \in [m_{L-1}].$$

Input   Layer 1 Layer 2 Layer 3 Layer 4

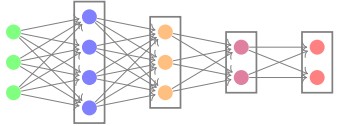

Figure 1: Standard Architecture

Input   Layer 1 Layer 2 Layer 3 Layer 4

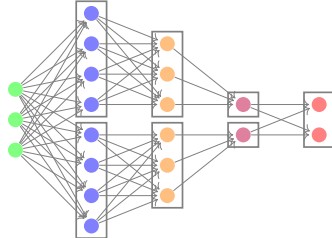

Figure 2: Parallel Architecture

To be specific, these two formulations have the same optimal value and the optimal solutions of one problem can be rescaled into the optimal solution of another solution. Based on the rescaling of parameters in $P_{\text{lin}}(t)$ , we characterize the dual problem of $P_{\text{lin}}(t)$ and its bi-dual, i.e., dual of the dual problem.

**Proposition 1** *The dual problem of $P_{\text{lin}}(t)$ is a convex optimization problem given by*

$$D_{\text{lin}}(t) = \max_{\mathbf{\Lambda}} \operatorname{tr}(\mathbf{\Lambda}^T \mathbf{Y})$$

$$s.t. \max_{\|\mathbf{W}_i\|_F \leq t, i \in [L-2], \|\mathbf{w}_{L-1}\|_2 \leq 1} \|\mathbf{\Lambda}^T \mathbf{X} \mathbf{W}_1 \dots \mathbf{W}_{L-2} \mathbf{w}_{L-1}\|_2 \leq 1.$$

*There exists a threshold of the number of branches $m^* \leq KN + 1$ such that $D_{\text{lin}}(t) = BD_{\text{lin}}(t)$, where $BD_{\text{lin}}(t)$ is the optimal value of the bi-dual problem*

$$BD_{\text{lin}}(t) = \min_{\{\mathbf{W}_{l,j}\}_{l \in [L], j \in [m^*]}} \sum_{j=1}^{m^*} \|\mathbf{w}_{L,j}^{\text{row}}\|_2,$$

$$s.t. \sum_{j=1}^{m^*} \mathbf{X} \mathbf{W}_{1,j} \dots \mathbf{W}_{L-2,j} \mathbf{w}_{L-1,j}^{\text{col}} \mathbf{w}_{L,j}^{\text{row}} = \mathbf{Y}, \tag{6}$$

$$\|\mathbf{W}_{i,j}\|_F \leq t, i \in [L-2], j \in [m^*], \|\mathbf{w}_{L-1,j}^{\text{col}}\|_2 \leq 1, j \in [m^*].$$

Detailed derivation of the dual and the bi-dual problems are provided in Appendix C.1. As $\mathbf{\Lambda} = 0$ is a strict feasible point for the dual problem, the optimal dual solutions exist due to classical results in strong duality for convex problems. The reason why we do not directly take the dual of $P_{\text{lin}}$ is that the objective function in $P_{\text{lin}}$ involves the weights of first $L-1$ layer, which prevents obtaining a non-trivial dual problem. An interesting observation is that the bi-dual problem is related to the minimum norm problem of a parallel neural network with balanced weights. Namely, the Frobenius norm of the weight matrices $\{\mathbf{W}_{l,j}\}_{l=1}^{L-2}$ in each branch $j \in [m]$ has the same upper bound $t$.

To calculate the value $P_{\text{lin}}(t)$ for fixed $t \in \mathbb{R}$, we introduce the definition of Schatten-$p$ norm.

**Definition 1** For a matrix $\mathbf{A} \in \mathbb{R}^{m \times n}$ and $p > 0$, the Schatten-$p$ quasi-norm of $\mathbf{A}$ is defined as

$$\|\mathbf{A}\|_{S_p} = \left( \sum_{i=1}^{\min\{m,n\}} \sigma_i^p(\mathbf{A}) \right)^{1/p},$$

where $\sigma_i(\mathbf{A})$ is the $i$-th largest singular value of $\mathbf{A}$.

The following proposition provides a closed-form solution for the sub-problem $P_{\text{lin}}(t)$ and determines its optimal value.

**Proposition 2** *Suppose that $\mathbf{W} \in \mathbb{R}^{d \times K}$ with rank $r$ is given. Assume that $m_l \geq r$ for $l = 1, \dots, L-1$. Consider the following optimization problem:*

$$\min_{\{\mathbf{W}_l\}_{l=1}^{L}} \frac{1}{2} \left( \|\mathbf{W}_1\|_F^2 + \dots + \|\mathbf{W}_L\|_F^2 \right), \ s.t. \ \mathbf{W}_1 \mathbf{W}_2 \dots \mathbf{W}_L = \mathbf{W}. \tag{7}$$

*Then, the optimal value of the problem (7) is given by $\frac{L}{2} \|\mathbf{W}\|_{S_{2/L}}^{2/L}$. Suppose that $\mathbf{W} = \mathbf{U} \mathbf{\Sigma} \mathbf{V}^T$. The optimal value is achieved when*

$$\mathbf{W}_l = \mathbf{U}_{l-1} \mathbf{\Sigma}^{1/L} \mathbf{U}_l^T, \quad i = l, \dots, L. \tag{8}$$

*Here $\mathbf{U}_0 = \mathbf{U}, \mathbf{U}_L = \mathbf{V}$ and for $l = 1, \dots, L-1$, $\mathbf{U}_l \in \mathbb{R}^{m_l \times r}$ satisfies that $\mathbf{U}_l^T \mathbf{U}_l = \mathbf{I}_r$.*

To the best of our knowledge, this result was not known previously. Proposition 2 implies that $P_{\text{lin}}$ can be equivalently written as

$$\min \frac{L}{2} \|\mathbf{W}\|_{S_{2/L}}^{2/L} \quad s.t. \ \mathbf{X} \mathbf{W} = \mathbf{Y}.$$

Denote $\mathbf{X}^\dagger$ as the pseudo inverse of $\mathbf{X}$. Although the objective is non-convex for $L \geq 3$, this problem has a closed-form solution as we show next.

**Theorem 2** *Suppose that $\mathbf{X}^\dagger \mathbf{Y} = \mathbf{U}\mathbf{\Sigma}\mathbf{V}^T$ is the singular value decomposition and let $r := rank(\mathbf{X}^\dagger \mathbf{Y})$. Assume that $m_l \geq r$ for $l = 1, \ldots, L - 1$. The optimal solution to $P_{\mathrm{lin}}$ is given in closed-form as follows:*

$$\mathbf{W}_l = \mathbf{U}_{l-1}\mathbf{\Sigma}^{1/L}\mathbf{U}_l^T, l \in [L] \tag{9}$$

*where $\mathbf{U}_0 = \mathbf{U}, \mathbf{U}_L = \mathbf{V}$. For $l = 1, \ldots, L - 1$, $\mathbf{U}_l \in \mathbb{R}^{m_l \times r}$ satisfies $\mathbf{U}_l^T \mathbf{U}_l = \mathbf{I}_r$.*

Based on Theorem 2, the optimal value of $P_{\mathrm{lin}}(t)$ and $D_{\mathrm{lin}}(t)$ can be precisely calculated as follows.

**Theorem 3** *Assume that $m_l \geq rank(\mathbf{X}^\dagger \mathbf{Y})$ for $l = 1, \ldots, L - 1$. For fixed $t > 0$, the optimal value of $P_{\mathrm{lin}}(t)$ and $D_{\mathrm{lin}}(t)$ are given by*

$$P_{\mathrm{lin}}(t) = t^{-(L-2)}\|\mathbf{X}^\dagger \mathbf{Y}\|_{S_{2/L}}, \tag{10}$$

*and*

$$D_{\mathrm{lin}}(t) = t^{-(L-2)}\|\mathbf{X}^\dagger \mathbf{Y}\|_*. \tag{11}$$

*Here $\|\cdot\|_*$ represents the nuclear norm. $P_{\mathrm{lin}}(t) = D_{\mathrm{lin}}(t)$ if and only if the singular values of $\mathbf{X}^\dagger \mathbf{Y}$ are equal.*

As a result, if the singular values of $\mathbf{X}^\dagger \mathbf{Y}$ are not equal to the same value, the duality gap exists, i.e., $P > D$, for standard deep linear networks with $L \geq 3$. We note that the optimal scale parameter $t$ for the primal problem $P_{\mathrm{lin}}$ is given by $t^* = \|\mathbf{W}^*\|_{S_{2/L}}^{1/L}$. This proves the first part of Theorem 1.

We conclude that, the deep linear network training problem has a duality gap whenever the depth is three or more. In contrast, there exists no duality gap for depth two. Nevertheless, the optimal solution can be obtained in closed form as we have shown. In the following section, we introduce a parallel multi-branch architecture that always has zero duality gap regardless of the depth.

## 3.2 Parallel deep linear neural networks

Now we consider the parallel multi-branch network structure as defined in Section 2, and consider the corresponding minimum norm optimization problem:

$$\min_{\{\mathbf{W}_{l,j}\}_{l \in [L], j \in [m]}} \frac{1}{2}\left(\sum_{l=1}^{L-1}\sum_{j=1}^{m}\|\mathbf{W}_{l,j}\|_F^2 + \|\mathbf{W}_L\|_F^2\right),$$

$$\text{s.t.} \sum_{j=1}^{m}\mathbf{X}\mathbf{W}_{1,j}\ldots\mathbf{W}_{L-2,j}\mathbf{w}_{L-1,j}^{\mathrm{col}}\mathbf{w}_{L,j}^{\mathrm{row}} = \mathbf{Y}. \tag{12}$$

Due to a rescaling to achieve the lower bound of the inequality of arithmetic and geometric means, we can formulate the problem (12) in the following way. In other words, two formulations (12) and (13) have the same optimal value and the optimal solutions of one problem can be mapped to the optimal solutions of another problem.

**Proposition 3** *The problem (12) can be formulated as*

$$\min_{\{\mathbf{W}_{l,j}\}_{l \in [L], j \in [m]}} \frac{L}{2}\sum_{j=1}^{m}\|\mathbf{w}_{L,j}^{\mathrm{row}}\|_2^{2/L},$$

$$s.t. \sum_{j=1}^{m}\mathbf{X}\mathbf{W}_{1,j}\ldots\mathbf{W}_{L-2,j}\mathbf{w}_{L-1,j}^{\mathrm{col}}\mathbf{w}_{L,j}^{\mathrm{row}} = \mathbf{Y}, \tag{13}$$

$$\|\mathbf{W}_{l,j}\|_F \leq 1, l \in [L-2], j \in [m], \|\mathbf{w}_{L-1,j}^{\mathrm{col}}\|_2 \leq 1, j \in [m].$$

We note that $z^{2/L}$ is a non-convex function of $z$ and we cannot hope to obtain a non-trivial dual. To solve this issue, we consider the $\|\cdot\|_F^L$ regularized objective given by

$$P_{\mathrm{lin}}^{\mathrm{prl}} = \min_{\{\mathbf{W}_{l,j}\}_{l \in [L], j \in [m]}} \frac{1}{2}\left(\sum_{l=1}^{L-1}\sum_{j=1}^{m}\|\mathbf{W}_{l,j}\|_F^L + \sum_{j=1}^{m}\|\mathbf{w}_{L,j}^{\mathrm{row}}\|_2^L\right),$$

$$\text{s.t.} \sum_{j=1}^{m}\mathbf{X}\mathbf{W}_{1,j}\ldots\mathbf{W}_{L-2,j}\mathbf{w}_{L-1,j}^{\mathrm{col}}\mathbf{w}_{L,j}^{\mathrm{row}} = \mathbf{Y}. \tag{14}$$

Utilizing the arithmetic and geometric mean (AM-GM) inequality, we can rescale the parameters and formulate (14). To be specific, the two formulations (14) and (15) have the same optimal value and the optimal solutions of one problem can be rescaled to the optimal solutions of another problem and vice versa.

**Proposition 4** *The problem* (14) *can be formulated as*

$$P_{\text{lin}}^{\text{prl}} = \min_{\{\mathbf{W}_{l,j}\}_{l \in [L], j \in [m]}} \frac{L}{2} \sum_{j=1}^{m} \|\mathbf{w}_{L,j}^{\text{row}}\|_2,$$

$$s.t. \sum_{j=1}^{m} \mathbf{X}\mathbf{W}_{1,j} \dots \mathbf{W}_{L-2,j} \mathbf{w}_{L-1,j}^{\text{col}} \mathbf{w}_{L,j}^{\text{row}} = \mathbf{Y}, \tag{15}$$

$$\|\mathbf{W}_{l,j}\|_F \leq 1, l \in [L-2], j \in [m], \|\mathbf{w}_{L-1,j}^{\text{col}}\|_2 \leq 1, j \in [m].$$

*The dual problem of $P_{\text{lin}}^{\text{prl}}$ is a convex problem*

$$D_{\text{lin}}^{\text{prl}} = \max_{\mathbf{\Lambda}} \text{tr}(\mathbf{\Lambda}^T \mathbf{Y}),$$

$$s.t. \max_{\|\mathbf{W}_i\|_F \leq 1, i \in [L-2], \|\mathbf{w}_{L-1}\|_2 \leq 1} \|\mathbf{\Lambda}^T \mathbf{X} \mathbf{W}_1 \dots \mathbf{W}_{L-2} \mathbf{w}_{L-1}\|_2 \leq L/2 \tag{16}$$

In contrary to the standard linear network model, the strong duality holds for the parallel linear network training problem (14).

**Theorem 4** *There exists a critical width $m^* \leq KN + 1$ such that as long as the number of branches $m \geq m^*$, the strong duality holds for the problem* (14). *Namely, $P_{\text{lin}}^{\text{prl}} = D_{\text{lin}}^{prl}$. The optimal values are both $\frac{L}{2}\|\mathbf{X}^\dagger \mathbf{Y}\|_*$.*

This implies that there exist equivalent convex problems which achieve the global optimum of the deep parallel linear network. Comparatively, optimizing deep parallel linear neural networks can be much easier than optimizing deep standard linear networks.

## 4 NEURAL NETWORKS WITH RELU ACTIVATION

### 4.1 STANDARD THREE-LAYER RELU NETWORKS

We first focus on the three-layer ReLU network with standard architecture. Specifically, we set $\phi(z) = \max\{z, 0\}$. Consider the minimum norm problem

$$P_{\text{ReLU}} = \min_{\{\mathbf{W}_i\}_{i=1}^3} \frac{1}{2} \sum_{i=1}^{3} \|\mathbf{W}_i\|_F^2, \text{ s.t. } ((\mathbf{X}\mathbf{W}_1)_+ \mathbf{W}_2)_+ \mathbf{W}_3 = \mathbf{Y}. \tag{17}$$

Here we denote $(z)_+ = \max\{z, 0\}$. Similarly, by introducing a scale parameter $t$, this problem can be formulated as $P_{\text{ReLU}} = \min_{t>0} \frac{1}{2}t^2 + P_{\text{ReLU}}(t)$, where $P_{\text{ReLU}}(t)$ is defined as

$$P_{\text{ReLU}}(t) = \min_{\{\mathbf{W}_i\}_{i=1}^3} \sum_{j=1}^{K} \|\mathbf{w}_{3,j}^{\text{row}}\|_2,$$

$$\text{s.t. } \|\mathbf{W}_1\|_F \leq t, \|\mathbf{w}_{2,j}^{\text{col}}\|_2 \leq 1, j \in [m_2], ((\mathbf{X}\mathbf{W}_1)_+ \mathbf{W}_2)_+ \mathbf{W}_3 = \mathbf{Y}. \tag{18}$$

The proof is analogous to the proof of Proposition 3.1 in (Ergen & Pilanci, 2021a). To be specific, these two formulations have the same optimal value and their optimal solutions can be mutually transformed into each other. For $\mathbf{W}_1 \in \mathbb{R}^{d \times m}$, we define the set

$$\mathcal{A}(\mathbf{W}_1) = \{((\mathbf{X}\mathbf{W}_1)_+ \mathbf{w}_2)_+ | \|\mathbf{w}_2\|_2 \leq 1\}. \tag{19}$$

We derive the convex dual problem of $P_{\text{ReLU}}(t)$ in the following proposition.

**Proposition 5** *The dual problem of $P_{\text{ReLU}}(t)$ defined in* (18) *is a convex problem defined as*

$$D_{\text{ReLU}}(t) = \max_{\mathbf{\Lambda}} \text{tr}(\mathbf{\Lambda}^T \mathbf{Y}), \text{ s.t. } \max_{\mathbf{W}_1: \|\mathbf{W}_1\|_F \leq t} \max_{\mathbf{v} \in \mathcal{A}(\mathbf{W}_1)} \|\mathbf{\Lambda}^T \mathbf{v}\|_2 \leq 1. \tag{20}$$

*There exists a threshold of the number of branches $m^* \leq KN + 1$ such that $D_{\mathrm{ReLU}}(t) = BD_{\mathrm{ReLU}}(t)$ where $BD_{\mathrm{ReLU}}(t)$ is the optimal value of the bi-dual problem*

$$BD_{\mathrm{ReLU}}(t) = \min_{\{\mathbf{W}_{1,j}\}_{j=1}^{m^*}, \mathbf{W}_2 \in \mathbb{R}^{m_1 \times m^*}, \mathbf{W}_3 \in \mathbb{R}^{m^* \times K}} \sum_{j=1}^{K} \|\mathbf{w}_{3,j}^{\mathrm{row}}\|_2,$$

$$s.t. \ \sum_{j=1}^{m^*} ((\mathbf{X}\mathbf{W}_{1,j})_+ \mathbf{w}_{2,j}^{\mathrm{col}})_+ \mathbf{w}_{3,j}^{\mathrm{row}} = \mathbf{Y}, \|\mathbf{W}_{1,j}\|_F \leq t, \|\mathbf{w}_{2,j}^{\mathrm{col}}\|_2 \leq 1, j \in [m^*].$$

(21)

We note that the bi-dual problem defined in (21) indeed optimizes with a parallel neural network satisfying $\|\mathbf{W}_{1,j}\|_F \leq t, \|\mathbf{w}_{2,j}^{\mathrm{col}}\|_2 \leq 1, j \in [m^*]$. For the case where the data matrix is with rank 1 and the neural network is with scalar output, we show that there is no duality gap. We extend the result in (Ergen & Pilanci, 2021d) from two-layer ReLU networks to three-layer ReLU networks.

**Theorem 5** *For a three-layer scalar-output ReLU network, let $\mathbf{X} = \mathbf{c}\mathbf{a}_0^T$ be a rank-one data matrix. Then, strong duality holds, i.e., $P_{\mathrm{ReLU}}(t) = D_{\mathrm{ReLU}}(t)$. Suppose that $\boldsymbol{\lambda}^*$ is the optimal solution to the dual problem $D_{\mathrm{ReLU}}(t)$, then the optimal weights for each layer can be formulated as*

$$\mathbf{W}_1 = t\,\mathrm{sign}(|(\boldsymbol{\lambda}^*)^T(\mathbf{c})_+| - |(\boldsymbol{\lambda}^*)^T(-\mathbf{c})_+|)\boldsymbol{\rho}_0\boldsymbol{\rho}_1^T, \mathbf{w}_2 = \boldsymbol{\rho}_1.$$

*Here $\boldsymbol{\rho}_0 = \mathbf{a}_0/\|\mathbf{a}_0\|_2$ and $\boldsymbol{\rho}_1 \in \mathbb{R}_+^{m_1}$ satisfies $\|\boldsymbol{\rho}_1\| = 1$.*

For general standard three-layer neural networks, although we have $BD_{\mathrm{ReLU}}(t) = D_{\mathrm{ReLU}}(t)$, it may not hold that $P_{\mathrm{ReLU}}(t) = D_{\mathrm{ReLU}}(t)$ as the bi-dual problem corresponds to optimizing a parallel neural network instead of a standard neural network to fit the labels.

To theoretically justify that the duality gap can be zero, we consider a parallel multi-branch architecture for ReLU networks in the next section.

## 4.2 PARALLEL DEEP RELU NETWORKS

For the corresponding parallel architecture, we show that there is no duality gap for arbitrary depth ReLU networks, as long as the number of branches is large enough. Consider the following minimum norm problem:

$$P_{\mathrm{ReLU}}^{\mathrm{prl}} = \min \frac{1}{2} \sum_{l=1}^{L-1} \sum_{j=1}^{m} \|\mathbf{W}_{l,j}\|_F^L + \|\mathbf{W}_L\|_F^L, \ \text{s.t.} \ \sum_{j=1}^{m} ((\mathbf{X}\mathbf{W}_{1,j})_+ \dots \mathbf{w}_{L-1,j}^{\mathrm{col}})_+ \mathbf{w}_{L,j}^{\mathrm{row}} = \mathbf{Y}.$$

(22)

As the ReLU activation is homogeneous, we can rescale the parameter to reformulate (22) and derive the dual problem. We note that two formulations (22) and (23) have the same optimal value and the optimal solutions of one problem can be rescaled to the optimal solutions of another problem and vice versa.

**Proposition 6** *The problem (22) can be reformulated as*

$$\min \frac{L}{2} \sum_{j=1}^{m} \|\mathbf{w}_{L,j}^{\mathrm{row}}\|_2,$$

$$s.t. \ \sum_{j=1}^{m} ((\mathbf{X}\mathbf{W}_{1,j})_+ \mathbf{w}_{L-1,j}^{\mathrm{col}})_+ \mathbf{w}_{L,j}^{\mathrm{row}} = \mathbf{Y}, \|\mathbf{W}_{l,j}\|_F \leq 1, l \in [L-2], \|\mathbf{w}_{L-1,j}^{\mathrm{col}}\|_2 \leq 1, j \in [m].$$

(23)

*The dual problem of (23) is a convex problem defined as*

$$D_{\mathrm{ReLU}}^{\mathrm{prl}} = \max \mathrm{tr}(\boldsymbol{\Lambda}^T \mathbf{Y}),$$

$$s.t. \ \max_{\mathbf{v} = ((\mathbf{X}\mathbf{W}_1)_+ \dots \mathbf{W}_{L-2})_+ \mathbf{w}_{L-1})_+, \|\mathbf{W}_l\|_F \leq 1, l \in [L-2], \|\mathbf{w}_{L-1}\|_2 \leq 1} \|\boldsymbol{\Lambda}^T \mathbf{v}\|_2 \leq L/2.$$

(24)

For deep parallel ReLU networks, we show that with sufficient number of parallel branches, the strong duality holds, i.e., $P = D$.

**Theorem 6** *Let $m^*$ be the threshold of the number of branches, which is upper bounded by $KN + 1$. Then, as long as the number of branches $m \geq m^*$, the strong duality holds for (23) in the sense that $P_{\text{ReLU}}^{\text{prl}} = D_{\text{ReLU}}^{\text{prl}}$.*

Similar to case of parallel deep linear networks, the parallel deep ReLU network also achieves zero-duality gap. Therefore, to find the global optimum for parallel deep ReLU network is equivalent to solve a convex program. This proves the second part of Theorem 1.

Based on the strong duality results, assuming that we can obtain an optimal solution to the convex dual problem (24), then we can construct an optimal solution to the primal problem (23) as follows.

**Theorem 7** *Let $\mathbf{\Lambda}^*$ be the optimal solution to (24). Denote the set of maximizers*

$$\arg\max_{\mathbf{v}=((\mathbf{XW}_1)_+ \ldots \mathbf{W}_{L-2})_+ \mathbf{w}_{L-1})_+, \|\mathbf{W}_l\|_F \leq 1, l \in [L-2], \|\mathbf{w}_{L-1}\|_2 \leq 1} \|(\mathbf{\Lambda}^*)^T \mathbf{v}\|_2 \tag{25}$$

*as $\{\mathbf{v}_1, \ldots, \mathbf{v}_{m^*}\}$, where $\mathbf{v}_i = ((\mathbf{XW}_{1,i})_+ \ldots \mathbf{W}_{L-2,i})_+ \mathbf{w}_{L-1,i})_+$ with $\|\mathbf{W}_{l,i}\|_F \leq 1, l \in [L-2]$ and $\|\mathbf{w}_{L-1,i}\|_2 \leq 1$ and $m^* \leq KN + 1$ is the critical threshold of the number of branches. Let $\mathbf{w}_{L,1}^{\text{row}}, \ldots, \mathbf{w}_{L,m^*}^{\text{row}}$ be an optimal solution to the convex problem*

$$P_{\text{ReLU}}^{\text{prl,sub}} = \min_{\mathbf{W}_L} \frac{L}{2} \sum_{j=1}^{m^*} \|\mathbf{w}_{L,j}^{\text{row}}\|_2, \ s.t. \ \sum_{j=1}^{m} ((\mathbf{XW}_{1,j})_+ \mathbf{w}_{L-1,j}^{\text{col}})_+ \mathbf{w}_{L,j}^{\text{row}} = \mathbf{Y}. \tag{26}$$

*Then, $(\mathbf{W}_1, \ldots, \mathbf{W}_L)$ is an optimal solution to (23).*

We note that finding the set of maximizers in (25) can be challenging in practice due to the high-dimensionality of the constraint set.

## 5 CONCLUSION

We present the convex duality framework for standard neural networks, considering both multi-layer linear networks and three-layer ReLU networks with rank-1. In stark contrast to the two-layer case, the duality gap can be non-zero for neural networks with depth three or more. Meanwhile, for neural networks with parallel architecture, with the regularization of $L$-th power of Frobenius norm in the parameters, we show that strong duality holds and the duality gap reduces to zero. A limitation of our work is that we primarily focus on minimum norm interpolation problems. We believe that our results can be easily generalized to a regularized training problems with general loss function, including squared loss, logistic loss, hinge loss, etc..

Another interesting research direction is investigating the complexity of solving our convex dual problems. Although the number of variables can be high for deep networks, the convex duality framework offers a rigorous theoretical perspective to the structure of optimal solutions. These problems can also shed light into the optimization landscape of their equivalent non-convex formulations. We note that it is not yet clear whether convex formulations of deep networks present practical gains in training. However, in Mishkin et al. (2022); Pilanci & Ergen (2020) it was shown that convex formulations provide significant computational speed-ups in training two-layer neural networks. Furthermore, similar convex analysis was also applied various architectures including batch normalization (Ergen et al., 2022b), vector output networks (Sahiner et al., 2021), threshold and polynomial activation networks (Ergen et al., 2023; Bartan & Pilanci, 2021), GANs (Sahiner et al., 2022a), autoregressive models (Gupta et al., 2021), and Transformers (Ergen et al., 2022a; Sahiner et al., 2022b).

### ACKNOWLEDGEMENTS

This work was partially supported by the National Science Foundation (NSF) under grants ECCS-2037304, DMS-2134248, NSF CAREER award CCF-2236829, the U.S. Army Research Office Early Career Award W911NF-21-1-0242, and the ACCESS – AI Chip Center for Emerging Smart Systems, sponsored by InnoHK funding, Hong Kong SAR.

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

# A    CONVEX DUALITY FOR TWO-LAYER NEURAL NETWORKS

We briefly review the convex duality theory for two-layer neural networks introduced in Ergen & Pilanci (2021a; 2020a). Consider the following weight-decay regularized training problem for a vector-output neural network architecture with $m$ hidden neurons

$$\min_{\mathbf{W}_1, \mathbf{W}_2} \frac{1}{2} \|\phi(\mathbf{X}\mathbf{W}_1)\mathbf{W}_2 - \mathbf{Y}\|_F^2 + \frac{\beta}{2}(\|\mathbf{W}_1\|_F^2 + \|\mathbf{W}_2\|_F^2), \tag{27}$$

where $\mathbf{W}_1 \in \mathbb{R}^{d \times m}$ and $\mathbf{W}_2 \in \mathbb{R}^{m \times K}$ are the variables, and $\beta > 0$ is a regularization parameter. Here $\phi$ is the activation function, which can be linear $\phi(z) = z$ or ReLU $\phi(z) = \max\{z, 0\}$. As long as the network is sufficiently overparameterized, there exists a feasible point for such that $\phi(\mathbf{X}\mathbf{W}_1)\mathbf{W}_2 = \mathbf{Y}$. Then, a minimum norm variant[1] of the training problem in (27) is given by

$$\min_{\mathbf{W}_1, \mathbf{W}_2} \frac{1}{2}(\|\mathbf{W}_1\|_F^2 + \|\mathbf{W}_2\|_F^2) \text{ s.t. } \phi(\mathbf{X}\mathbf{W}_1)\mathbf{W}_2 = \mathbf{Y}. \tag{28}$$

As shown in Pilanci & Ergen (2020), after a suitable rescaling, this problem can be reformulated as

$$\min_{\mathbf{W}_1, \mathbf{W}_2} \sum_{j=1}^m \|\mathbf{w}_{2,j}^{\text{row}}\|_2, \text{ s.t. } \quad \phi(\mathbf{X}\mathbf{W}_1)\mathbf{W}_2 = \mathbf{Y}, \|\mathbf{w}_{1,j}^{\text{col}}\|_2 \leq 1, j \in [m]. \tag{29}$$

where $[m] = \{1, \dots, m\}$. Here $\mathbf{w}_{2,j}^{\text{row}}$ represents the $j$-th row of $\mathbf{W}_2$ and $\mathbf{w}_{1,j}^{\text{col}}$ denotes the $j$-th column of $\mathbf{W}_1$. The rescaling does not change the solution to (28). By taking the dual with respect to $\mathbf{W}_1$ and $\mathbf{W}_2$, the dual problem of (29) with respect to variables is a convex optimization problem given by

$$\max_{\boldsymbol{\Lambda}} \text{tr}(\boldsymbol{\Lambda}^T \mathbf{Y}), \text{ s.t. } \max_{\mathbf{u}:\|\mathbf{u}\|_2 \leq 1} \|\boldsymbol{\Lambda}^T \phi(\mathbf{X}\mathbf{u})\|_2 \leq 1, \tag{30}$$

where $\boldsymbol{\Lambda} \in \mathbb{R}^{N \times K}$ is the dual variable. Provided that $m \geq m^*$, where $m^*$ is a critical threshold of width upper bounded by $m^* \leq N + 1$, the strong duality holds, i.e., the optimal value of the primal problem (29) equals to the optimal value of the dual problem (30).

# B    DEEP LINEAR NETWORKS WITH GENERAL LOSS FUNCTIONS

We consider deep linear networks with general loss functions, i.e.,

$$\min_{\{\mathbf{W}_l\}_{l=1}^L} \ell(\mathbf{X}\mathbf{W}_1 \dots \mathbf{W}_L, \mathbf{Y}) + \frac{\beta}{2} \sum_{i=1}^L \|\mathbf{W}_i\|_F^2,$$

where $\ell(\mathbf{Z}, \mathbf{Y})$ is a general loss function and $\beta > 0$ is a regularization parameter. According to Proposition 2, the above problem is equivalent to

$$\min_{\mathbf{W}} \ell(\mathbf{X}\mathbf{W}, \mathbf{Y}) + \frac{\beta L}{2} \|\mathbf{W}\|_{S_{2/L}}^{2/L}. \tag{31}$$

The $\ell_2$ regularization term becomes the Schatten-$2/L$ quasi-norm on $\mathbf{W}$ to the power $2/L$. Suppose that there exists $\mathbf{W}$ such that $l(\mathbf{X}\mathbf{W}, \mathbf{Y}) = 0$. With $\beta \to 0$, asymptotically, the optimal solution to the problem (31) converges to the optimal solution of

$$\min_{\mathbf{W}} \|\mathbf{W}\|_{S_{2/L}}^{2/L}, \text{ s.t. } \ell(\mathbf{X}\mathbf{W}, \mathbf{Y}) = 0. \tag{32}$$

In other words, the $\ell_2$ regularization explicitly regularizes the training problem to find a low-rank solution $\mathbf{W}$.

---

[1]This corresponds to weak regularization, i.e., $\beta \to 0$ in (27) as considered in Wei et al. (2018).

## C    PROOFS OF MAIN RESULTS FOR LINEAR NETWORKS

### C.1    PROOF OF PROPOSITION 1

Consider the Lagrangian function

$$L(\mathbf{W}_1, \ldots, \mathbf{W}_L, \mathbf{\Lambda}) = \sum_{j=1}^{K} \|\mathbf{w}_{L,j}\|_2 + \mathrm{tr}(\mathbf{\Lambda}^T(\mathbf{Y} - \mathbf{X}\mathbf{W}_1 \ldots \mathbf{W}_L)). \tag{33}$$

Here $\Lambda \in \mathbb{R}^{N \times K}$ is the dual variable. We note that

$$\begin{aligned}
P(t) &= \min_{\mathbf{W}_1, \ldots, \mathbf{W}_L} \max_{\mathbf{\Lambda}} L(\mathbf{W}_1, \ldots, \mathbf{W}_L, \mathbf{\Lambda}), \\
&\quad \text{s.t. } \|\mathbf{W}_i\|_F \leq t, i \in [L-2], \|\mathbf{w}_{L-1,j}^{\mathrm{col}}\|_2 \leq 1, j \in [m_{L-1}], \\
&= \min_{\mathbf{W}_1, \ldots, \mathbf{W}_{L-1}} \max_{\mathbf{\Lambda}} \mathrm{tr}(\mathbf{\Lambda}^T\mathbf{Y}) - \sum_{j=1}^{m_{L-1}} \mathbb{I}\left(\|\mathbf{\Lambda}^T\mathbf{X}\mathbf{W}_1 \ldots \mathbf{W}_{L-2}\mathbf{w}_{L-1,j}\|_2 \leq 1\right), \\
&\quad \text{s.t. } \|\mathbf{W}_i\|_F \leq t, i \in [L-2], \|\mathbf{w}_{L-1,j}^{\mathrm{col}}\|_2 \leq 1, j \in [m_{L-1}], \\
&= \min_{\mathbf{W}_1, \ldots, \mathbf{W}_{L-2}, \mathbf{W}_{L-1}} \max_{\mathbf{\Lambda}} \mathrm{tr}(\mathbf{\Lambda}^T\mathbf{Y}) - \mathbb{I}\left(\|\mathbf{\Lambda}^T\mathbf{X}\mathbf{W}_1 \ldots \mathbf{W}_{L-2}\mathbf{w}_{L-1}\|_2 \leq 1\right), \\
&\quad \text{s.t. } \|\mathbf{W}_i\|_F \leq t, i \in [L-2], \|\mathbf{w}_{L-1}\|_2 \leq 1.
\end{aligned} \tag{34}$$

Here $\mathbb{I}(A)$ is 0 if the statement $A$ is true. Otherwise it is $+\infty$. For fixed $\mathbf{W}_1, \ldots, \mathbf{W}_{L-1}$, the constraint on $\mathbf{W}_L$ is linear so we can exchange the order of $\max_{\mathbf{\Lambda}}$ and $\min_{\mathbf{W}_L}$ in the second line of (34).

By exchanging the order of min and max, we obtain the dual problem

$$\begin{aligned}
D(t) &= \max_{\mathbf{\Lambda}} \min_{\mathbf{W}_1, \ldots, \mathbf{W}_{L-2}} \mathrm{tr}(\mathbf{\Lambda}^T\mathbf{Y}) - \mathbb{I}\left(\|\mathbf{\Lambda}^T\mathbf{X}\mathbf{W}_1 \ldots \mathbf{W}_{L-2}\mathbf{w}_{L-1}\|_2 \leq 1\right), \\
&\quad \text{s.t. } \|\mathbf{W}_i\|_F \leq t, i \in [L-2], \|\mathbf{w}_{L-1}\|_2 \leq 1, \\
&= \max_{\mathbf{\Lambda}} \mathrm{tr}(\mathbf{\Lambda}^T\mathbf{Y}) \\
&\quad \text{s.t. } \|\mathbf{\Lambda}^T\mathbf{X}\mathbf{W}_1 \ldots \mathbf{W}_{L-2}\mathbf{w}_{L-1}\|_2 \leq 1 \\
&\quad \forall \|\mathbf{W}_i\|_F \leq t, i \in [L-2], \|\mathbf{w}_{L-1}\|_2 \leq 1.
\end{aligned} \tag{35}$$

Now we derive the bi-dual problem. The dual problem can be reformulated as

$$\begin{aligned}
&\max_{\mathbf{\Lambda}} \mathrm{tr}(\mathbf{\Lambda}^T\mathbf{Y}), \\
&\text{s.t. } \|\mathbf{\Lambda}^T\mathbf{X}\mathbf{W}_1 \ldots \mathbf{W}_{L-2}\mathbf{w}_{L-1}\|_2 \leq 1, \\
&\qquad \forall (\mathbf{W}_1, \ldots, \mathbf{W}_{L-2}, \mathbf{w}_{L-1}) \in \Theta.
\end{aligned} \tag{36}$$

Here the set $\Theta$ is defined as

$$\Theta = \{(\mathbf{W}_1, \ldots, \mathbf{W}_{L-2}, \mathbf{w}_{L-1}) | \|\mathbf{W}_i\|_F \leq t, i \in [L-2], \|\mathbf{w}_{L-1}\|_2 \leq 1\}. \tag{37}$$

By writing $\theta = (\mathbf{W}_1, \ldots, \mathbf{W}_{L-2}, \mathbf{w}_{L-1})$, the dual of the problem (36) is given by

$$\begin{aligned}
&\min \|\boldsymbol{\mu}\|_{\mathrm{TV}}, \\
&\text{s.t. } \int_{\theta \in \Theta} \mathbf{X}\mathbf{W}_1 \ldots \mathbf{W}_{L-2}\mathbf{w}_{L-1} d\boldsymbol{\mu}(\theta) = \mathbf{Y}.
\end{aligned} \tag{38}$$

Here $\boldsymbol{\mu} : \Sigma \to \mathbb{R}^K$ is a signed vector measure and $\Sigma$ is a $\sigma$-field of subsets of $\Theta$. The norm $\|\boldsymbol{\mu}\|_{\mathrm{TV}}$ is the total variation of $\boldsymbol{\mu}$, which can be calculated by

$$\|\boldsymbol{\mu}\|_{TV} = \sup_{u: \|u(\theta)\|_2 \leq 1} \left\{ \int_\Theta u^T(\theta) d\mu(\theta) =: \sum_{i=1}^{K} \int_\Theta u_i(\theta) d\mu_i(\theta) \right\}, \tag{39}$$

where we write $\boldsymbol{\mu} = \begin{bmatrix} \mu_1 \\ \vdots \\ \mu_K \end{bmatrix}$. The formulation in (38) has infinite width in each layer. According to Theorem 10 in Appendix G, the measure $\boldsymbol{\mu}$ in the integral can be represented by finitely many Dirac delta functions. Therefore, we can rewrite the problem (38) as

$$
\begin{aligned}
&\min \sum_{j=1}^{m^*} \|\mathbf{w}_{L,j}^{\text{row}}\|_2, \\
&\text{s.t. } \sum_{j=1}^{m^*} \mathbf{X}\mathbf{W}_{1,j}\ldots\mathbf{W}_{L-2,j}\mathbf{w}_{L-1,j}^{\text{col}}\mathbf{w}_{L,j}^{\text{row}} = \mathbf{Y}, \\
&\|\mathbf{W}_{i,j}\|_F \le t, i \in [L-2], \|\mathbf{w}_{L-1,j}^{\text{col}}\|_2 \le 1, j \in [m^*].
\end{aligned}
\tag{40}
$$

Here the variables are $\mathbf{W}_{i,j}$ for $i \in [L-2]$ and $j \in [m^*]$, $\mathbf{W}_{L-1}$ and $\mathbf{W}_L$. As the strong duality holds for the problem (40) and (36), we can reformulate the problem of $D_{\text{lin}}(t)$ as the bi-dual problem (40).

## C.2 PROOF OF PROPOSITION 2

We restate Proposition 2 with details.

**Proposition 7** *Suppose that $\mathbf{W} \in \mathbb{R}^{d \times K}$ with rank $r$ is given. Consider the following optimization problem:*

$$
\min \frac{1}{2}\left(\|\mathbf{W}_1\|_F^2 + \cdots + \|\mathbf{W}_L\|_F^2\right), \text{ s.t. } \mathbf{W}_1\mathbf{W}_2\ldots\mathbf{W}_L = \mathbf{W},
\tag{41}
$$

*in variables $\mathbf{W}_i \in \mathbb{R}^{m_{i-1} \times m_i}$. Here $m_0 = d$, $m_L = K$ and $m_i \ge r$ for $i = 1, \ldots, L-1$. Then, the optimal value of the problem (41) is given by*

$$
\frac{L}{2}\|\mathbf{W}\|_{S_{2/L}}^{2/L}.
\tag{42}
$$

*Suppose that $\mathbf{W} = \mathbf{U}\boldsymbol{\Sigma}\mathbf{V}^T$. The optimal value can be achieved when*

$$
\mathbf{W}_i = \mathbf{U}_{i-1}\boldsymbol{\Sigma}^{1/L}\mathbf{U}_i^T, \quad i = 1, \ldots, N, \mathbf{U}_0 = \mathbf{U}, \mathbf{U}_L = \mathbf{V}.
\tag{43}
$$

*Here $\mathbf{U}_i \in \mathbb{R}^{r \times m_i}$ satisfies that $\mathbf{U}_i^T\mathbf{U}_i = I$.*

We start with two lemmas.

**Lemma 1** *Suppose that $A \in \mathbb{S}^{n \times n}$ is a positive semi-definite matrix. Then, for any $0 < p < 1$, we have*

$$
\sum_{i=1}^{n} A_{ii}^p \ge \sum_{i=1}^{n} \lambda_i(A)^p.
\tag{44}
$$

*Here $\lambda_i$ is the $i$-th largest eigenvalue of $A$.*

**Lemma 2** *Suppose that $P \in \mathbb{R}^{d \times d}$ is a projection matrix, i.e., $P^2 = P$. Then, for arbitrary $W \in \mathbb{R}^{d \times K}$, we have*

$$
\sigma_i(PW) \le \sigma_i(W),
$$

*where $\sigma_i(W)$ represents the $i$-th largest singular value of $W$.*

Now, we present the proof for Proposition 2. For $L = 1$, the statement apparently holds. Suppose that for $L = l$ this statement holds. For $L = l+1$, by writing $\mathbf{A} = \mathbf{W}_2\ldots\mathbf{W}_{l+1}$, we have

$$
\begin{aligned}
&\min \|\mathbf{W}_1\|_F^2 + \cdots + \|\mathbf{W}_L\|_F^2, \text{ s.t. } \mathbf{W}_1\mathbf{W}_2\ldots\mathbf{W}_{l+1} = \mathbf{W} \\
&= \min \|\mathbf{W}_1\|_F^2 + l\|\mathbf{A}\|_{2/l}^{2/l}, \text{ s.t. } \mathbf{W}_1\mathbf{A} = \mathbf{W}, \\
&= \min t^2 + l\|\mathbf{A}\|_{2/l}^{2/l}, \text{ s.t. } \mathbf{W}_1\mathbf{A} = \mathbf{W}, \|\mathbf{W}_1\|_F \le t.
\end{aligned}
\tag{45}
$$

Suppose that $t$ is fixed. It is sufficient to consider the following problem:

$$
\min \|\mathbf{A}\|_{2/l}^{2/l}, \text{ s.t. } \mathbf{W}_1\mathbf{A} = \mathbf{W}, \|\mathbf{W}_1\|_F \le t.
\tag{46}
$$

Suppose that there exists $\mathbf{W}_1$ and $\mathbf{A}$ such that $\mathbf{W} = \mathbf{W}_1 \mathbf{A}$. Then, we have $\mathbf{W}\mathbf{A}^\dagger \mathbf{A} = \mathbf{W}_1 \mathbf{A}\mathbf{A}^\dagger \mathbf{A} = \mathbf{W}$. As $\mathbf{W}\mathbf{A}^\dagger = \mathbf{W}_1 \mathbf{A}\mathbf{A}^\dagger$, according to Lemma 2, $\|\mathbf{W}\mathbf{A}^\dagger\|_F \leq \|\mathbf{W}_1\|_F \leq t$. Therefore, $(\mathbf{W}\mathbf{A}^\dagger, \mathbf{A})$ is also feasible for the problem (46). Hence, the problem (46) is equivalent to

$$\min \|\mathbf{A}\|_{2/l}^{2/l}, \text{ s.t. } \mathbf{W}\mathbf{A}^\dagger \mathbf{A} = \mathbf{W}, \|\mathbf{W}\mathbf{A}^\dagger\|_F \leq t. \tag{47}$$

Assume that $\mathbf{W}$ is with rank $r$. Suppose that $\mathbf{A} = \mathbf{U}\boldsymbol{\Sigma}\mathbf{V}^T$, where $\boldsymbol{\Sigma} \in \mathbb{R}^{r_0 \times r_0}$. Here $r_0 \geq r$. Then, we have $\mathbf{A}^\dagger = \mathbf{V}\boldsymbol{\Sigma}^{-1}\mathbf{U}^T$. We note that

$$\begin{aligned} &\|\mathbf{W}\mathbf{A}^\dagger\|_F^2 \\ &= \text{tr}(\mathbf{W}\mathbf{V}\boldsymbol{\Sigma}^{-2}\mathbf{V}^T\mathbf{W}^T) \\ &= \text{tr}(\mathbf{V}^T\mathbf{W}^T\mathbf{W}\mathbf{V}\boldsymbol{\Sigma}^{-2}) \end{aligned} \tag{48}$$

Denote $G(\mathbf{V}) = \mathbf{V}^T\mathbf{W}^T\mathbf{W}\mathbf{V}$. This implies that

$$\sum_{i=1}^{r} \sigma_i(\mathbf{A})^{-2} (G(\mathbf{V}))_{ii} \leq t^2.$$

Therefore, we have

$$\left(\sum_{i=1}^{r_0} \sigma_i(\mathbf{A})^{-2} (G(\mathbf{V}))_{ii}\right) \left(\sum_{i=1}^{r_0} \sigma_i(\mathbf{A})^{2/l}\right)^l \geq \left(\sum_{i=1}^{r_0} (G(\mathbf{V}))_{ii}^{1/(l+1)}\right)^{l+1}.$$

As $\mathbf{W}\mathbf{V}^T\mathbf{V} = \mathbf{W}$, the non-zero eigenvalues of $G(\mathbf{V})$ are exactly the non-zero eigenvalues of $\mathbf{W}\mathbf{V}\mathbf{V}^T\mathbf{W}^T = \mathbf{W}\mathbf{W}^T$, i.e., the square of non-zero singular values of $\mathbf{W}$. From Lemma 1, we have

$$\sum_{i=1}^{r_0} (G(\mathbf{V}))_{ii}^{1/(l+1)} \geq \sum_{i=1}^{r_0} \lambda_i(G(\mathbf{V}))^{1/(l+1)} \geq \sum_{i=1}^{r} \sigma_i(\mathbf{W})^{2/(l+1)}. \tag{49}$$

Therefore, we have

$$\|\mathbf{A}\|_{S_{2/l}}^{2/l} = \sum_{i=1}^{r_0} \sigma_i(\mathbf{A})^{2/l} \geq t^{-2/l} \left(\sum_{i=1}^{r} \sigma_i(\mathbf{W})^{2/(l+1)}\right)^{(l+1)/l} \tag{50}$$

This also implies that

$$\begin{aligned} &\min \|\mathbf{A}\|_{2/l}^{2/l}, \text{ s.t. } \mathbf{W}_1 \mathbf{A} = \mathbf{W}, \|\mathbf{W}_1\|_F \leq t \\ &\geq t^{-2/l} \left(\sum_{i=1}^{r} \sigma_i(\mathbf{W})^{2/(l+1)}\right)^{(l+1)/l}. \end{aligned} \tag{51}$$

Suppose that $\mathbf{W} = \sum_{i=1}^{r} u_i \sigma_i v_i^T$ is the SVD of $\mathbf{W}$. We can let

$$\begin{aligned} \mathbf{A} &= \frac{\left(\sum_{i=1}^{r} \sigma_i^{2/(l+1)}\right)^{1/2}}{t} \sum_{i=1}^{r} u_i \sigma_i^{l/(l+1)} \rho_i^T, \\ \mathbf{W}_1 &= \frac{t}{\left(\sum_{i=1}^{r} \sigma_i^{2/(l+1)}\right)^{1/2}} \sum_{i=1}^{r} \rho_i \sigma_i^{1/(l+1)} v_i^T. \end{aligned} \tag{52}$$

Here $\|\rho_i\|_2 = 1$ and $\rho_i^T \rho_j = 0$ for $i \neq j$. Then, $\mathbf{W}_1 \mathbf{A} = \mathbf{W}$ and $\|\mathbf{W}_1\|_F \leq t$. We also have

$$\begin{aligned} \|\mathbf{A}\|_{S_{2/L}}^{2/L} &= t^{-2/l} \left(\sum_{i=1}^{r} \sigma_i^{2/(l+1)}\right)^{1/l} \sum_{i=1}^{r} \sigma_i^{2/(l+1)} \\ &= t^{-2/l} \left(\sum_{i=1}^{r} \sigma_i(\mathbf{W})^{2/(l+1)}\right)^{(l+1)/l}. \end{aligned}$$

In summary, we have

$$\min t^2 + l\|\mathbf{A}\|_{2/l}^{S_{2/l}}, \text{ s.t. } \mathbf{W}_1\mathbf{A} = \mathbf{W}, \|\mathbf{W}_1\|_F \le t.$$

$$=\min_{t>0} t^2 + lt^{-2/l}\left(\sum_{i=1}^r \sigma_i(\mathbf{W})^{2/(l+1)}\right)^{(l+1)/l}$$

$$=(l+1)\left(\sum_{i=1}^r \sigma_i(\mathbf{W})^{2/(l+1)}\right)^{(l+1)/2} \tag{53}$$

$$=\|\mathbf{W}\|_{S_{2/(l+1)}}^{2/(l+1)}.$$

This completes the proof.

### C.3 PROOF OF THEOREM 2

From Proposition 2, the minimum norm problem (5) is equivalent to

$$\min L\|\mathbf{W}\|_{S_{2/L}}^{2/L}, \text{ s.t. } \mathbf{X}\mathbf{W} = \mathbf{Y}, \tag{54}$$

in variable $\mathbf{W} \in \mathbb{R}^{d \times K}$. According to Lemma 2, for any feasible $\mathbf{W}$ satisfying $\mathbf{X}\mathbf{W} = \mathbf{Y}$, because $\mathbf{X}^\dagger \mathbf{X}\mathbf{W} = \mathbf{X}^\dagger \mathbf{Y}$ and $\mathbf{X}^\dagger \mathbf{X}$ is a projection matrix, we have

$$L\|\mathbf{W}\|_{S_{2/L}}^{2/L} \ge L\|\mathbf{X}^\dagger \mathbf{Y}\|_{S_{2/L}}^{2/L}. \tag{55}$$

We also note that $\mathbf{X}\mathbf{X}^\dagger \mathbf{Y} = \mathbf{X}\mathbf{X}^\dagger \mathbf{X}\mathbf{W} = \mathbf{X}\mathbf{W} = \mathbf{Y}$. Therefore, $\mathbf{X}^\dagger \mathbf{Y}$ is also feasible for the problem (54). This indicates that $P_{\text{lin}} = \frac{L}{2}\|\mathbf{X}^\dagger \mathbf{Y}\|_{S_{2/L}}^{2/L}$.

### C.4 PROOF OF THEOREM 3

For a feasible point $(\mathbf{W}_1, \ldots, \mathbf{W}_L)$ for $P_{\text{lin}}(t)$, we note that $(\mathbf{W}_1/t, \ldots, \mathbf{W}_{L-2}/t, \mathbf{W}_{L-1}, t^{L-2}\mathbf{W}_L)$ is feasible for $P_{\text{lin}}(1)$. This implies that $t^{L-2}P_{\text{lin}}(t) = P_{\text{lin}}(1)$, or equivalently, $P_{\text{lin}}(t) = t^{-(L-2)}P_{\text{lin}}(1)$. Recall that

$$P_{\text{lin}} = \min_{t>0} \frac{L-2}{2}t^2 + t^{-(L-2)}P_{\text{lin}}(1)$$

$$=\frac{L}{2}\left(P_{\text{lin}}(1)\right)^{2/L}. \tag{56}$$

From Theorem 2, we have $P_{\text{lin}} = \frac{L}{2}\|\mathbf{X}^\dagger \mathbf{Y}\|_{S_{2/L}}^{2/L}$. This implies that $P_{\text{lin}}(1) = \|\mathbf{X}^\dagger \mathbf{Y}\|_{S_{2/L}}$ and

$$P_{\text{lin}}(t) = t^{-(L-2)}\|\mathbf{X}^\dagger \mathbf{Y}\|_{S_{2/L}}. \tag{57}$$

For the dual problem $D_{\text{lin}}(t)$ defined in (35), we note that

$$\|\mathbf{\Lambda}^T\mathbf{X}\mathbf{W}_1 \ldots \mathbf{W}_{L-2}\mathbf{w}_{L-1}\|_2$$

$$\le\|\mathbf{\Lambda}^T\mathbf{X}\mathbf{W}_1 \ldots \mathbf{W}_{L-2}\|_2\|\mathbf{w}_{L-1}\|_2$$

$$\le\|\mathbf{\Lambda}^T\mathbf{X}\|_2 \prod_{l=1}^{L-2} \|\mathbf{W}_l\|_2\|\mathbf{w}_{L-1}\|_2 \tag{58}$$

$$\le\|\mathbf{\Lambda}^T\mathbf{X}\|_2 \prod_{l=1}^{L-2} \|\mathbf{W}_l\|_F\|\mathbf{w}_{L-1}\|_2 = t^{L-2}\|\mathbf{\Lambda}^T\mathbf{X}\|_2.$$

The equality can be achieved when $\mathbf{W}_l = t\mathbf{u}_l\mathbf{u}_{l+1}^T$ for $l \in [L-2]$, where $\|\mathbf{u}_l\|_2 = 1$ for $l = 1, \ldots, L-1$. Specifically, we set $\mathbf{u}_{L-1} = \mathbf{w}_{L-1}$ and let $\mathbf{u}_0$ as right singular vector corresponds to the largest singular value of $\mathbf{\Lambda}^T\mathbf{X}$. Therefore, the constraints on $\mathbf{\Lambda}$ is equivalent to

$$\|\mathbf{\Lambda}^T\mathbf{X}\|_2 \le t^{-(L-2)}. \tag{59}$$

Thus, according to the Von Neumann's trace inequality, it follows

$$\mathrm{tr}(\mathbf{\Lambda}^T \mathbf{Y}) = \mathrm{tr}(\mathbf{\Lambda}^T \mathbf{X} \mathbf{X}^\dagger \mathbf{Y}) \le \|\mathbf{\Lambda}^T \mathbf{X}\|_2 \|\mathbf{X}^\dagger \mathbf{Y}\|_* \le t^{-(L-2)} \|\mathbf{X}^\dagger \mathbf{Y}\|_*. \tag{60}$$

Suppose that $\mathbf{X}^\dagger \mathbf{Y} = \mathbf{U} \mathbf{\Sigma} \mathbf{V}^T$ is the singular value decomposition. Let $\mathbf{\Sigma} = \mathbf{diag}(\sigma_1, \dots, \sigma_r)$ where $\sigma_1 \ge \sigma_2 \ge \cdots \ge \sigma_r > 0$ and $r = \mathrm{rank}(\mathbf{X}^\dagger \mathbf{Y})$. We note that

$$
\begin{aligned}
\|\mathbf{X}^\dagger \mathbf{Y}\|_{S_{2/L}} &= \left( \sum_{i=1}^r \sigma_i^{2/L} \right)^{L/2} \\
&= \sigma_1 \left( 1 + \sum_{i=2}^r (\sigma_i/\sigma_1)^{2/L} \right)^{L/2} \\
&\ge \sigma_1 \left( 1 + \sum_{i=2}^r (\sigma_i/\sigma_1) \right) \\
&= \sum_{i=1}^r \sigma_r.
\end{aligned}
\tag{61}
$$

The equality holds if and only if $\sigma_1 = \cdots = \sigma_r$. This is because for given $x \in (0,1)$ and $a \ge 1$, $(a + x^p)^{1/p}$ is strictly decreasing w.r.t. $p \in (0,1]$. As a result, we have $D_{\mathrm{lin}}(t) = t^{-(L-2)} \|\mathbf{X}^\dagger \mathbf{Y}\|_* \le t^{-(L-2)} \|\mathbf{X}^\dagger \mathbf{Y}\|_{S_{2/L}} = P_{\mathrm{lin}}(t)$. The equality is achieved if and only if the singular values of $\mathbf{X}^\dagger \mathbf{Y}$ are the same. In other words, the inequality is strict when $\mathbf{X}^\dagger \mathbf{Y}$ has different singular values. Then, the duality gap exists for the standard neural network.

## C.5 PROOF OF PROPOSITION 3

For simplicity, we write $\mathbf{W}_{L-1,j} = \mathbf{w}_{L-1,j}^{\mathrm{col}}$ and $\mathbf{W}_{L,j} = \mathbf{w}_{L,j}^{\mathrm{row}}$ for $j \in [m]$. For the $j$-th branch of the parallel network, let $\hat{\mathbf{W}}_{l,j} = \alpha_{l,j} \mathbf{W}_{l,j}$ for $l \in [L]$. Here $\alpha_{l,j} > 0$ for $l \in [L]$ and they satisfies that $\prod_{l=1}^L \alpha_{l,j} = 1$ for $j \in [m]$. Therefore, we have

$$\mathbf{X} \mathbf{W}_{1,j} \dots \mathbf{W}_{L-2,j} \mathbf{w}_{L-1,j}^{\mathrm{col}} \mathbf{w}_{L,j}^{\mathrm{row}} = \mathbf{X} \hat{\mathbf{W}}_{1,j} \dots \hat{\mathbf{W}}_{L-2,j} \hat{\mathbf{w}}_{L-1,j}^{\mathrm{col}} \hat{\mathbf{w}}_{L,j}^{\mathrm{row}}. \tag{62}$$

This implies that $\{\hat{\mathbf{W}}_{l,j}\}_{l \in [L], j \in [m]}$ is also feasible for the problem (12). According to the the inequality of arithmetic and geometric means, the objective function in (12) is lower bounded by

$$
\begin{aligned}
& \frac{1}{2} \sum_{j=1}^m \sum_{l=1}^L \alpha_{l,j}^2 \|\mathbf{W}_{l,j}\|_F^2 \\
&\ge \sum_{j=1}^m \frac{L}{2} \prod_{l=1}^L \left( \alpha_{l,j}^{2/L} \|\mathbf{W}_{l,j}\|_F^{2/L} \right) \\
&= \frac{L}{2} \sum_{j=1}^m \prod_{l=1}^L \|\mathbf{W}_{l,j}\|_F^{2/L}.
\end{aligned}
\tag{63}
$$

The equality is achieved when $\alpha_{l,j} = \frac{\prod_{l=1}^L \|\mathbf{W}_{l,j}\|_F^{1/L}}{\|\mathbf{W}_{l,j}\|_F}$ for $l \in [L]$ and $j \in [m]$. As the scaling operation does not change $\prod_{l=1}^L \|\mathbf{W}_{l,j}\|_F^{2/L}$, we can simply let $\|\mathbf{W}_{l,j}\|_F = 1$ and the lower bound becomes $\frac{L}{2} \sum_{i=1}^m \|\mathbf{W}_{L,j}\|_F^{2/L} = \frac{L}{2} \sum_{i=1}^m \|\mathbf{w}_{L,j}^{\mathrm{row}}\|_2^{2/L}$. This completes the proof.

## C.6 PROOF OF PROPOSITION 4

We first show that the problem (14) is equivalent to (15). The proof is analogous to the proof of Proposition 3. For simplicity, we write $\mathbf{W}_{L-1,j} = \mathbf{w}_{L-1,j}^{\mathrm{col}}$ and $\mathbf{W}_{L,j} = \mathbf{w}_{L,j}^{\mathrm{row}}$ for $j \in [m]$. Let $\alpha_{l,j} > 0$ for $l \in [L]$ and they satisfies that $\prod_{l=1}^L \alpha_{l,j} = 1$ for $j \in [m]$. Consider another parallel

network $\{\hat{\mathbf{W}}_{l,j}\}_{l\in[L],j\in[m]}$ whose $j$-th branch is defined by $\hat{\mathbf{W}}_{l,j} = \alpha_{l,j}\mathbf{W}_{l,j}$ for $l \in [L]$. As $\prod_{l=1}^{L}\alpha_{l,j} = 1$, we have

$$\mathbf{X}\mathbf{W}_{1,j}\ldots\mathbf{W}_{L-2,j}\mathbf{w}_{L-1,j}^{\text{col}}\mathbf{w}_{L,j}^{\text{row}} = \mathbf{X}\hat{\mathbf{W}}_{1,j}\ldots\hat{\mathbf{W}}_{L-2,j}\hat{\mathbf{w}}_{L-1,j}^{\text{col}}\hat{\mathbf{w}}_{L,j}^{\text{row}}. \tag{64}$$

This implies that $\{\hat{\mathbf{W}}_{l,j}\}_{l\in[L],j\in[m]}$ is also feasible for the problem (14). According to the the inequality of arithmetic and geometric means, the objective function in (12) is lower bounded by

$$\frac{1}{2}\sum_{j=1}^{m}\sum_{l=1}^{L}\alpha_{l,j}^{L}\|\mathbf{W}_{l,j}\|_{F}^{L}$$

$$\geq \sum_{j=1}^{m}\frac{L}{2}\prod_{l=1}^{L}(\alpha_{l,j}\|\mathbf{W}_{l,j}\|_{F}) \tag{65}$$

$$= \frac{L}{2}\sum_{j=1}^{m}\prod_{l=1}^{L}\|\mathbf{W}_{l,j}\|_{F}.$$

The equality is achieved when $\alpha_{l,j} = \frac{\prod_{l=1}^{L}\|\mathbf{W}_{l,j}\|_{F}^{1/L}}{\|\mathbf{W}_{l,j}\|_{F}}$ for $l \in [L]$ and $j \in [m]$. As the scaling operation does not change $\prod_{l=1}^{L}\|\mathbf{W}_{l,j}\|_{F}$, we can simply let $\|\mathbf{W}_{l,j}\|_{F} = 1$ and the lower bound becomes $\frac{L}{2}\sum_{i=1}^{m}\|\mathbf{W}_{L,j}\|_{F} = \frac{L}{2}\sum_{i=1}^{m}\|\mathbf{w}_{L,j}^{\text{row}}\|_{2}$. Hence, the problem (14) is equivalent to (15).

For the problem (15), we consider the Lagrangian function

$$L(\mathbf{W}_{1},\ldots,\mathbf{W}_{L}) = \frac{L}{2}\sum_{j=1}^{m}\|\mathbf{w}_{L,j}^{\text{row}}\|_{2} + \text{tr}\left(\mathbf{\Lambda}^{T}(\mathbf{Y} - \sum_{j=1}^{m}\mathbf{X}\mathbf{W}_{1,j}\ldots\mathbf{W}_{L-1,j}^{\text{col}}\mathbf{W}_{L,j}^{\text{row}})\right). \tag{66}$$

The primal problem is equivalent to

$$P_{\text{lin}}^{\text{prl}} = \min_{\mathbf{W}_{1},\ldots,\mathbf{W}_{L}}\max_{\mathbf{\Lambda}}L(\mathbf{W}_{1},\ldots,\mathbf{W}_{L},\mathbf{\Lambda}),$$

$$\text{s.t. } \|\mathbf{W}_{l,j}\|_{F} \leq t, j \in [m_{l}], l \in [L-2], \|\mathbf{w}_{L-1,j}^{\text{col}}\|_{2} \leq 1, j \in [m_{L-1}],$$

$$= \min_{\mathbf{W}_{1},\ldots,\mathbf{W}_{L-1}}\max_{\mathbf{\Lambda}}\min_{\mathbf{W}_{L}}L(\mathbf{W}_{1},\ldots,\mathbf{W}_{L},\mathbf{\Lambda}),$$

$$\text{s.t. } \|\mathbf{W}_{l,j}\|_{F} \leq 1, l \in [L-2], \|\mathbf{w}_{L-1,j}^{\text{col}}\|_{2} \leq 1, j \in [m], \tag{67}$$

$$= \min_{\mathbf{W}_{1},\ldots,\mathbf{W}_{L-1}}\max_{\mathbf{\Lambda}}\text{tr}(\mathbf{\Lambda}^{T}\mathbf{Y}) - \sum_{j=1}^{m_{L}}\mathbb{I}\left(\|\mathbf{\Lambda}^{T}\mathbf{X}\mathbf{W}_{1,j}\ldots\mathbf{W}_{L-2,j}\mathbf{w}_{L-1,j}^{\text{col}}\|_{2} \leq L/2\right),$$

$$\text{s.t. } \|\mathbf{W}_{l,j}\|_{F} \leq 1, l \in [L-2], \|\mathbf{w}_{L-1,j}^{\text{col}}\|_{2} \leq 1, j \in [m].$$

The dual problem follows

$$D_{\text{lin}}^{\text{prl}} = \max_{\mathbf{\Lambda}}\text{tr}(\mathbf{\Lambda}^{T}\mathbf{Y}),$$

$$\text{s.t. } \|\mathbf{\Lambda}^{T}\mathbf{X}\mathbf{W}_{1,j}\ldots\mathbf{W}_{L-2,j}\|_{2} \leq L/2,$$

$$\forall\|\mathbf{W}_{l,j}\|_{F} \leq 1, l \in [L-2], \|\mathbf{W}_{L-1,j}^{\text{col}}\|_{2} \leq 1, j \in [m], \tag{68}$$

$$= \max_{\mathbf{\Lambda}}\text{tr}(\mathbf{\Lambda}^{T}\mathbf{Y}),$$

$$\text{s.t. } \|\mathbf{\Lambda}^{T}\mathbf{X}\mathbf{W}_{1}\ldots\mathbf{W}_{L-2}\mathbf{w}_{L-1}\|_{2} \leq L/2,$$

$$\forall\|\mathbf{W}_{i}\|_{F} \leq 1, i \in [L-2], \|\mathbf{w}_{L-1}\|_{2} \leq 1.$$

### C.7 PROOF OF THEOREM 4

We can rewrite the dual problem as

$$D_{\text{lin}}^{\text{prl}} = \max_{\mathbf{\Lambda}}\text{tr}(\mathbf{\Lambda}^{T}\mathbf{Y}),$$

$$\text{s.t. } \|\mathbf{\Lambda}^{T}\mathbf{X}\mathbf{W}_{1}\ldots\mathbf{W}_{L-2}\mathbf{w}_{L-1}\|_{2} \leq L/2, \tag{69}$$

$$\forall(\mathbf{W}_{1},\ldots,\mathbf{W}_{L-2},\mathbf{w}_{L-1}) \in \Theta,$$

where the set $\Theta$ is defined as

$$\Theta = \{(\mathbf{W}_1, \ldots, \mathbf{W}_{L-2}, \mathbf{w}_{L-1}) | \|\mathbf{W}_l\|_F \leq 1, l \in [L-2], \|\mathbf{w}_{L-1}\|_2 \leq 1\}. \tag{70}$$

By writing $\theta = (\mathbf{W}_1, \ldots, \mathbf{W}_{L-2}, \mathbf{w}_{L-1})$, the bi-dual problem, i.e., the dual problem of (69), is given by

$$\begin{aligned} \min &\ \|\boldsymbol{\mu}\|_{\text{TV}}, \\ \text{s.t.} &\ \int_{\theta \in \Theta} \mathbf{X} \mathbf{W}_1 \ldots \mathbf{W}_{L-2} \mathbf{w}_{L-1} d\boldsymbol{\mu}(\theta) = \mathbf{Y}. \end{aligned} \tag{71}$$

Here $\boldsymbol{\mu} : \Sigma \to \mathbb{R}^K$ is a signed vector measure, where $\Sigma$ is a $\sigma$-field of subsets of $\Theta$ and $\|\boldsymbol{\mu}\|_{\text{TV}}$ is its total variation. The formulation in (71) has infinite width in each layer. According to Theorem 10 in Appendix G, the measure $\boldsymbol{\mu}$ in the integral can be represented by finitely many Dirac delta functions. Therefore, there exists a critical threshold of the number of branchs $m^* < KN + 1$ such that we can rewrite the problem (71) as

$$\min \sum_{j=1}^{m^*} \|\mathbf{w}_{L,j}^{\text{row}}\|_2,$$

$$\text{s.t.} \sum_{j=1}^{m^*} \mathbf{X} \mathbf{W}_{1,j} \ldots \mathbf{W}_{L-2,j} \mathbf{w}_{L-1,j}^{\text{col}} \mathbf{w}_{L,j}^{\text{row}} = \mathbf{Y}, \tag{72}$$

$$\|\mathbf{W}_{i,j}\|_F \leq 1, l \in [L-2], \|\mathbf{w}_{L-1,j}^{\text{col}}\|_2 \leq 1, j \in [m^*].$$

Here the variables are $\mathbf{W}_{l,j}$ for $l \in [L-2]$ and $j \in [m^*]$, $\mathbf{W}_{L-1}$ and $\mathbf{W}_L$. This is equivalent to (15). As the strong duality holds for the problem (69) and (71), the primal problem (15) is equivalent to the dual problem (69) as long as $m \geq m^*$.

Now, we compute the optimal value of $D_{\text{lin}}^{\text{prl}}$. Similar to the proof of Theorem 3, we can show that the constraints in the dual problem (69) is equivalent to

$$\|\boldsymbol{\Lambda}^T \mathbf{X}\|_2 \leq L/2. \tag{73}$$

Therefore, we have

$$\text{tr}(\boldsymbol{\Lambda}^T \mathbf{Y}) \leq \|\boldsymbol{\lambda}^T \mathbf{X}\|_2 \|\mathbf{X}^\dagger \mathbf{Y}\|_* \leq \frac{L}{2} \|\mathbf{X}^\dagger \mathbf{Y}\|_*. \tag{74}$$

This implies that $P_{\text{lin}}^{\text{prl}} = D_{\text{lin}}^{\text{prl}} = \frac{L}{2} \|\mathbf{X}^\dagger \mathbf{Y}\|_*$.

## D  STAIRS OF DUALITY GAP FOR STANDARD DEEP LINEAR NETWORKS

We consider partially dualizing the non-convex optimization problem by exchanging a subset of the minimization problems with respect to the hidden layers. Consider the Lagrangian for the primal problem of standard deep linear network

$$\begin{aligned} P_{\text{lin}}(t) = \min_{\{\mathbf{W}_l\}_{l=1}^{L-1}} \max_{\boldsymbol{\Lambda}} &\ \text{tr}(\boldsymbol{\Lambda}^T \mathbf{Y}) - \mathbb{I}\left(\|\boldsymbol{\Lambda}^T \mathbf{X} \mathbf{W}_1 \ldots \mathbf{W}_{L-2} \mathbf{w}_{L-1}\|_2 \leq 1\right), \\ \text{s.t.} &\ \|\mathbf{W}_i\|_F \leq t, i \in [L-2], \|\mathbf{w}_{L-1}\|_2 \leq 1. \end{aligned} \tag{75}$$

By changing the order of $L-2$ mins and the max in (75), for $l = 0, 1, \ldots, L-2$, we can define the $l$-th partial "dual" problem

$$\begin{aligned} D_{\text{lin}}^{(l)}(t) = \min_{\mathbf{W}_1, \ldots \mathbf{W}_l} \max_{\boldsymbol{\Lambda}} \min_{\mathbf{W}_{l+1}, \ldots, \mathbf{W}_{L-2}} &\ \text{tr}(\boldsymbol{\Lambda}^T \mathbf{Y}) - \mathbb{I}\left(\|\boldsymbol{\Lambda}^T \mathbf{X} \mathbf{W}_1 \ldots \mathbf{W}_{L-2} \mathbf{w}_{L-1}\|_2 \leq 1\right), \\ \text{s.t.} &\ \|\mathbf{W}_i\|_F \leq t, i \in [L-2], \|\mathbf{w}_{L-1}\|_2 \leq 1. \end{aligned} \tag{76}$$

For $l = 0$, $D_{\text{lin}}^{(l)}(t)$ corresponds the primal problem $P_{\text{lin}}(t)$, while for $l = L-2$, $D_{\text{lin}}^{(l)}(t)$ is the dual problem $D_{\text{lin}}(t)$. From the following proposition, we illustrate that the dual problem of $D_{\text{lin}}^{(l)}(t)$ corresponds to a minimum norm problem of a neural network with parallel structure.

**Proposition 8** *There exists a threshold of the number of branches $m^* \leq KN + 1$ such that the problem $D_{\text{lin}}^{(l)}(t)$ is equivalent to the "bi-dual" problem*

$$\min \sum_{j=1}^{m^*} \|\mathbf{w}_{L,j}^{\text{row}}\|_2,$$

$$s.t. \sum_{j=1}^{m^*} \mathbf{X}\mathbf{W}_1 \dots \mathbf{W}_l \mathbf{W}_{l+1,j} \dots \mathbf{W}_{L-2,j} \mathbf{w}_{L-1,j}^{\text{col}} \mathbf{w}_{L,j}^{\text{row}} = \mathbf{Y}, \tag{77}$$

$$\|\mathbf{W}_i\|_F \leq t, i \in [l], \|\mathbf{W}_{i,j}\|_F \leq t, i = l+1, \dots, L-2, j \in [m^*],$$

$$\|\mathbf{w}_{L-1,j}^{\text{col}}\|_2 \leq 1, j \in [m^*],$$

*where the variables are $\mathbf{W}_i \in \mathbb{R}^{m_{i-1} \times m_i}$ for $i \in [l]$, $\mathbf{W}_{i,j} \in \mathbb{R}^{m_{i-1} \times m_i}$ for $i = l+1, \dots, L-2$, $j \in [m^*]$, $\mathbf{W}_{L-1} \in \mathbb{R}^{m_{L-2} \times m^*}$ and $\mathbf{W}_L \in \mathbb{R}^{m^* \times m_L}$.*

We can interpret the problem (77) as the minimum norm problem of a linear network with parallel structures in $(l+1)$-th to $(L-2)$-th layers. This indicates that for $l = 0, 1, \dots, L-2$, the bi-dual formulation of $D_{\text{lin}}^{(l)}(t)$ can be viewed as an interpolation from a network with standard structure to a network with parallel structure. Now, we calculate the exact value of $D_{\text{lin}}^{(l)}(t)$.

**Proposition 9** *The optimal value $D_{\text{lin}}^{(l)}(t)$ follows*

$$D_{\text{lin}}^{(l)}(t) = t^{-(L-2)} \|\mathbf{X}^\dagger \mathbf{Y}\|_{S_{2/(l+2)}}. \tag{78}$$

Suppose that the eigenvalues $\mathbf{X}^\dagger \mathbf{Y}$ are not identical to each other. Then, we have

$$P_{\text{lin}}(t) = D_{\text{lin}}^{(L-2)}(t) > D_{\text{lin}}^{(L-3)}(t) > \dots > D_{\text{lin}}^{(0)}(t) = D(t). \tag{79}$$

In Figure 3, we plot $D_{\text{lin}}^{(l)}(t)$ for $l = 0, \dots, 5$ for an example.

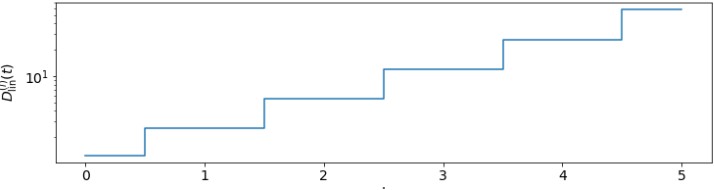

Figure 3: Example of $D_{\text{lin}}^{(l)}(t)$.

### D.1 PROOF OF PROPOSITION 9

We note that

$$\max_{\mathbf{\Lambda}} \text{tr}(\mathbf{\Lambda}^T \mathbf{Y}),$$

$$s.t. \|\mathbf{\Lambda}^T \mathbf{X}\mathbf{W}_1 \dots \mathbf{W}_{L-2}\|_2 \leq 1, \|\mathbf{W}_i\|_F \leq t, i = l+1, \dots, L-2, \tag{80}$$

$$= \max_{\mathbf{\Lambda}} \min_{\mathbf{W}_{j+1},\dots,\mathbf{W}_{L-2}} \text{tr}(\mathbf{\Lambda}^T \mathbf{Y}), s.t. \|\mathbf{\Lambda}^T \mathbf{X}\mathbf{W}_1 \dots \mathbf{W}_l\|_2 \leq t^{-(L-2-l)}.$$

Therefore, we can rewrite $D_{\text{lin}}^{(l)}(t)$ as

$$D_{\text{lin}}^{(l)}(t) = \min_{\mathbf{W}_1,\dots \mathbf{W}_l} \max_{\mathbf{\Lambda}} \text{tr}(\mathbf{\Lambda}^T \mathbf{Y}),$$

$$s.t. \|\mathbf{\Lambda}^T \mathbf{X}\mathbf{W}_1 \dots \mathbf{W}_l\|_2 \leq t^{-(L-2-l)}, \|\mathbf{W}_i\|_F \leq t, i \in [l],$$

$$= \min_{\mathbf{W}_1,\dots \mathbf{W}_l} \max_{\mathbf{\Lambda}} t^{-(L-2-l)} \text{tr}(\mathbf{\Lambda}^T \mathbf{Y}), \tag{81}$$

$$s.t. \|\mathbf{\Lambda}^T \mathbf{X}\mathbf{W}_1 \dots \mathbf{W}_l\|_2 \leq 1, \|\mathbf{W}_i\|_F \leq t, i \in [l].$$

From the equation (10), we note that

$$
\begin{aligned}
&\min_{\mathbf{W}_1,\dots\mathbf{W}_j} \max_{\mathbf{\Lambda}} \operatorname{tr}(\mathbf{\Lambda}^T\mathbf{Y}) \\
&\text{s.t. } \|\mathbf{\Lambda}^T\mathbf{X}\mathbf{W}_1\dots\mathbf{W}_j\|_2 \leq 1, \|\mathbf{W}_i\|_F \leq t, i \in [j], \\
&= \min \sum_{j=1}^{K} \|\mathbf{w}_{l+2,j}\|_2, \\
&\text{s.t. } \|\mathbf{W}_i\|_F \leq t, i \in [L-2], \|\mathbf{w}_{L-1,j}\|_2 \leq 1, j \in [m_{L-1}], \\
&\qquad \mathbf{X}\mathbf{W}_1\dots\mathbf{W}_{l+2} = \mathbf{Y} \\
&= t^{-l}\|\mathbf{X}^\dagger\mathbf{Y}\|_{S_{2/(l+2)}}.
\end{aligned}
\tag{82}
$$

This completes the proof.

# E  PROOFS OF MAIN RESULTS FOR ReLU NETWORKS

## E.1  PROOF OF PROPOSITION 5

For the problem of $P(t)$, introduce the Lagrangian function

$$
L(\mathbf{W}_1,\mathbf{W}_2,\mathbf{W}_3,\mathbf{\Lambda}) = \sum_{j=1}^{K} \|\mathbf{w}_{3,j}^{\text{row}}\|_2 - \operatorname{tr}(\mathbf{\Lambda}^T(((\mathbf{X}\mathbf{W}_1)_+\mathbf{W}_2)_+\mathbf{W}_3 - \mathbf{Y})).
\tag{83}
$$

According to the convex duality of two-layer ReLU network, we have

$$
\begin{aligned}
P_{\text{ReLU}}(t) &= \min_{\|\mathbf{W}_1\|_F \leq t, \|\mathbf{w}_2\| \leq 1} \max_{\mathbf{\Lambda}} \operatorname{tr}(\mathbf{\Lambda}^T\mathbf{Y}) - \mathbb{I}(\|\mathbf{\Lambda}^T((\mathbf{X}\mathbf{W}_1)_+\mathbf{w}_2)_+\|_2 \leq 1) \\
&= \min_{\|\mathbf{W}_1\|_F \leq t} \max_{\mathbf{\Lambda}} \min_{\|\mathbf{w}_2\| \leq 1} \operatorname{tr}(\mathbf{\Lambda}^T\mathbf{Y}) - \mathbb{I}(\|\mathbf{\Lambda}^T((\mathbf{X}\mathbf{W}_1)_+\mathbf{w}_2)_+\|_2 \leq 1) \\
&= \min_{\|\mathbf{W}_1\|_F \leq t} \max_{\mathbf{\Lambda}} \operatorname{tr}(\mathbf{\Lambda}^T\mathbf{Y}), \text{ s.t. } \|\mathbf{\Lambda}^T\mathbf{v}\|_2 \leq 1, \forall \mathbf{v} \in \mathcal{A}(\mathbf{W}_1).
\end{aligned}
\tag{84}
$$

By changing the min and max, we obtain the dual problem.

$$
D_{\text{ReLU}}(t) = \max_{\mathbf{\Lambda}} \operatorname{tr}(\mathbf{\Lambda}^T\mathbf{Y}), \text{ s.t. } \|\mathbf{\Lambda}^T\mathbf{v}\|_2 \leq 1, \mathbf{v} \in \mathcal{A}(\mathbf{W}_1), \forall\|\mathbf{W}_1\|_F \leq t.
\tag{85}
$$

The dual of the dual problem writes

$$
\begin{aligned}
&\min \|\boldsymbol{\mu}\|_{\text{TV}}, \\
&\text{s.t. } \int_{\|\mathbf{W}_1\|_F \leq t, \|\mathbf{w}_2\|_2 \leq 1} ((\mathbf{X}\mathbf{W}_1)_+\mathbf{w}_2)_+ \, d\boldsymbol{\mu}(\mathbf{W}_1,\mathbf{w}_2) = \mathbf{Y}.
\end{aligned}
\tag{86}
$$

Here $\boldsymbol{\mu}$ is a signed vector measure and $\|\boldsymbol{\mu}\|_{\text{TV}}$ is its total variation. Similar to the proof of Proposition 1, we can find a finite representation for the optimal measure and transform this problem to

$$
\begin{aligned}
&\min_{\{\mathbf{W}_{1,j}\}_{j=1}^{m^*}, \mathbf{W}_2 \in \mathbb{R}^{m_1 \times m^*}, \mathbf{W}_3 \in \mathbb{R}^{m^* \times K}} \sum_{j=1}^{K} \|\mathbf{w}_{3,j}\|_2, \\
&\text{s.t. } \sum_{j=1}^{m^*} ((\mathbf{X}\mathbf{W}_{1,j})_+\mathbf{w}_{2,j})_+\mathbf{w}_{3,j}^T = \mathbf{Y}, \|\mathbf{W}_{1,j}\|_F \leq t, \|\mathbf{w}_{2,j}\|_2 \leq 1.
\end{aligned}
\tag{87}
$$

Here $m^* \leq KN + 1$. This completes the proof.

## E.2  PROOF OF THEOREM 5

For rank-1 data matrix that $\mathbf{X} = \mathbf{c}\mathbf{a}_0^T$, suppose that $\mathbf{A}_1 = (\mathbf{X}\mathbf{W}_1)_+$. It is easy to observe that

$$
\mathbf{A}_1 = (\mathbf{c})_+\mathbf{a}_{1,+}^T + (-\mathbf{c})_+\mathbf{a}_{1,-}^T,
$$

Here we let $\mathbf{a}_{1,+} = (\mathbf{W}_1^T \mathbf{a}_0)_+$ and $\mathbf{a}_{1,-} = (-\mathbf{W}_1^T \mathbf{a}_0)_+$.

For a three-layer network, suppose that $\boldsymbol{\lambda}^*$ is the optimal solution to the dual problem $D_{\text{ReLU}}(t)$. We consider the extreme points defined by

$$\underset{\|\mathbf{W}_1\|_F \leq t, \|\mathbf{w}_2\|_2 \leq 1}{\arg\max} |(\boldsymbol{\lambda}^*)^T ((\mathbf{X}\mathbf{W}_1)_+ \mathbf{w}_2)_+|. \tag{88}$$

For fixed $\mathbf{W}_1$, because $\mathbf{a}_{1,+}^T \mathbf{a}_{1,-} = 0$, suppose that

$$\mathbf{w}_2 = u_1 \mathbf{a}_{1,+} + u_2 \mathbf{a}_{1,-} + u_3 \mathbf{r},$$

where $\mathbf{r}^T \mathbf{a}_{1,+} = \mathbf{r}^T \mathbf{a}_{1,-} = 0$ and $\|\mathbf{r}\|_2 = 1$. The maximization problem on $\mathbf{w}_2$ reduces to

$$\underset{u_1, u_2, u_3}{\arg\max} \left| (\boldsymbol{\lambda}^*)^T (\mathbf{c})_+ \|\mathbf{a}_{1,+}\|_2^2 (u_1)_+ + (\boldsymbol{\lambda}^*)^T (-\mathbf{c})_+ \|\mathbf{a}_{1,-}\|_2^2 (u_2)_+ \right|$$
$$\text{s.t. } u_1^2 \|\mathbf{a}_{1,+}\|_2^2 + u_2^2 \|\mathbf{a}_{1,+}\|_2^2 + u_3^2 \leq 1.$$

If $(\boldsymbol{\lambda}^*)^T (\mathbf{c})_+$ and $(\boldsymbol{\lambda}^*)^T (-\mathbf{c})_+$ have different signs, then the optimal value is

$$\max\{|(\boldsymbol{\lambda}^*)^T (\mathbf{c})_+| \|\mathbf{a}_{1,+}\|_2, |(\boldsymbol{\lambda}^*)^T (-\mathbf{c})_+| \|\mathbf{a}_{1,-}\|_2\}.$$

And the corresponding optimal $\mathbf{w}_2$ is $\mathbf{w}_2 = \mathbf{a}_{1,+}/\|\mathbf{a}_{1,+}\|_2$ or $\mathbf{w}_2 = \mathbf{a}_{1,-}/\|\mathbf{a}_{1,-}\|_2$. Then, the problem becomes

$$\underset{\mathbf{W}_1}{\arg\max} \max\{|(\boldsymbol{\lambda}^*)^T (\mathbf{c})_+| \|\mathbf{a}_{1,+}\|_2, |(\boldsymbol{\lambda}^*)^T (-\mathbf{c})_+| \|\mathbf{a}_{1,-}\|_2\}.$$

We note that

$$\max\{\|\mathbf{a}_{1,+}\|_2, \|\mathbf{a}_{1,-}\|_2\} \leq \|\mathbf{W}_1^T \mathbf{a}_0\|_2 \leq \|\mathbf{W}_1\|_2 \|\mathbf{a}_0\|_2 \leq t\|\mathbf{a}_0\|_2.$$

Thus the optimal $\mathbf{W}_1$ is given by

$$\mathbf{W}_1 = t\,\text{sign}(|(\boldsymbol{\lambda}^*)^T (\mathbf{c})_+| - |(\boldsymbol{\lambda}^*)^T (-\mathbf{c})_+|) \boldsymbol{\rho}_0 \boldsymbol{\rho}_1^T.$$

Here $\boldsymbol{\rho}_0 = \mathbf{a}_0/\|\mathbf{a}_0\|_2$ and $\boldsymbol{\rho}_1 \in \mathbb{R}_+^{m_l}$ satisfies $\|\boldsymbol{\rho}_1\| = 1$. This implies that the optimal $\mathbf{w}_2$ is given by $\mathbf{w}_2 = \boldsymbol{\rho}_1$.

On the other hand, if $(\boldsymbol{\lambda}^*)^T (\mathbf{c})_+$ and $(\boldsymbol{\lambda}^*)^T (-\mathbf{c})_+$ have same signs, then, the optimal $\mathbf{w}_2$ follows

$$\mathbf{w}_2 = \frac{|(\boldsymbol{\lambda}^*)^T (\mathbf{c})_+| \mathbf{a}_{1,+} + |(\boldsymbol{\lambda}^*)^T (-\mathbf{c})_+| \mathbf{a}_{1,-}}{\sqrt{((\boldsymbol{\lambda}^*)^T (\mathbf{c})_+)^2 \|\mathbf{a}_{1,+}\|_2^2 + ((\boldsymbol{\lambda}^*)^T (-\mathbf{c})_+)^2 \|\mathbf{a}_{1,-}\|_2^2}}.$$

The maximization problem of $\mathbf{W}_1$ is equivalent to

$$\underset{\|\mathbf{W}_1\|_F \leq t}{\arg\max} ((\boldsymbol{\lambda}^*)^T (\mathbf{c})_+)^2 \|\mathbf{a}_{1,+}\|_2^2 + ((\boldsymbol{\lambda}^*)^T (\mathbf{c})_-)^2 \|\mathbf{a}_{1,-}\|_2^2.$$

By noting that

$$\|\mathbf{a}_{1,+}\|_2^2 + \|\mathbf{a}_{1,-}\|_2^2 = \|\mathbf{W}_1^T \mathbf{a}_0\|_2^2 \leq \|\mathbf{W}_1\|_2^2 \|\mathbf{a}_0\|_2^2 \leq t^2 \|\mathbf{a}_0\|_2^2,$$

the optimal $\mathbf{W}_1$ is given by

$$\mathbf{W}_1 = t\,\text{sign}(|(\boldsymbol{\lambda}^*)^T (\mathbf{c})_+| - |(\boldsymbol{\lambda}^*)^T (-\mathbf{c})_+|) \boldsymbol{\rho}_0 \boldsymbol{\rho}_1^T.$$

Here $\boldsymbol{\rho}_0 = \mathbf{a}_0/\|\mathbf{a}_0\|_2$ and $\boldsymbol{\rho}_1 \in \mathbb{R}_+^{m_1}$ satisfies $\|\boldsymbol{\rho}_1\| = 1$.

### E.3 PROOF OF PROPOSITION 6

Analogous to the proof of Proposition 4, we can reformulate (22) into (23). The rest of the proof is analogous to the proof of Proposition 4. For the problem (23), we consider the Lagrangian function

$$L(\mathbf{W}_1, \ldots, \mathbf{W}_L)$$
$$= \frac{L}{2} \sum_{j=1}^{m} \|\mathbf{w}_{L,j}^{\text{row}}\|_2 + \text{tr}\left( \boldsymbol{\Lambda}^T (\mathbf{Y} - \sum_{j=1}^{m} (((\mathbf{X}\mathbf{W}_{1,j})_+ \cdots \cdots \mathbf{W}_{L-2,j})_+ \mathbf{w}_{L-1,j}^{\text{col}})_+ \mathbf{w}_{L,j}^{\text{row}}) \right). \tag{89}$$

The primal problem is equivalent to

$$
\begin{aligned}
&P_{\text{ReLU}}^{\text{prl}} \\
&= \min_{\mathbf{W}_1,\ldots,\mathbf{W}_L} \max_{\mathbf{\Lambda}} L(\mathbf{W}_1,\ldots,\mathbf{W}_L,\mathbf{\Lambda}), \\
&\quad \text{s.t. } \|\mathbf{W}_{l,j}\|_F \le t, j \in [m_l], l \in [L-2], \|\mathbf{w}_{L-1,j}^{\text{col}}\|_2 \le 1, j \in [m_{L-1}], \\
&= \min_{\mathbf{W}_1,\ldots,\mathbf{W}_{L-1}} \max_{\mathbf{\Lambda}} \min_{\mathbf{W}_L} L(\mathbf{W}_1,\ldots,\mathbf{W}_L,\mathbf{\Lambda}), \\
&\quad \text{s.t. } \|\mathbf{W}_{l,j}\|_F \le 1, l \in [L-2], \|\mathbf{w}_{L-1,j}^{\text{col}}\|_2 \le 1, j \in [m], \\
&= \min_{\mathbf{W}_1,\ldots,\mathbf{W}_{L-1}} \max_{\mathbf{\Lambda}} \operatorname{tr}(\mathbf{\Lambda}^T \mathbf{Y}) - \sum_{j=1}^m \mathbb{I}\left( \|\mathbf{\Lambda}^T(((\mathbf{X}\mathbf{W}_{1,j})_+ \ldots \mathbf{W}_{L-2,j})_+ \mathbf{w}_{L-1,j}^{\text{col}})_+\|_2 \le L/2 \right), \\
&\quad \text{s.t. } \|\mathbf{W}_{l,j}\|_F \le 1, l \in [L-2], \|\mathbf{w}_{L-1,j}^{\text{col}}\|_2 \le 1, j \in [m].
\end{aligned}
\tag{90}
$$

By exchanging the order of $\min$ and $\max$, the dual problem follows

$$
\begin{aligned}
D_{\text{ReLU}}^{\text{prl}} &= \max_{\mathbf{\Lambda}} \operatorname{tr}(\mathbf{\Lambda}^T \mathbf{Y}), \\
&\quad \text{s.t. } \|\mathbf{\Lambda}^T(((\mathbf{X}\mathbf{W}_{1,j})_+ \ldots \mathbf{W}_{L-2,j})_+ \mathbf{w}_{L-1,j}^{\text{col}})_+\|_2 \le L/2, \\
&\qquad \forall \|\mathbf{W}_{l,j}\|_F \le 1, l \in [L-2], \|\mathbf{w}_{L-1,j}^{\text{col}}\|_2 \le 1, j \in [m], \\
&= \max_{\mathbf{\Lambda}} \operatorname{tr}(\mathbf{\Lambda}^T \mathbf{Y}), \\
&\quad \text{s.t. } \|\mathbf{\Lambda}^T(((\mathbf{X}\mathbf{W}_1)_+ \ldots \mathbf{W}_{L-2})_+ \mathbf{w}_{L-1})_+\|_2 \le L/2, \\
&\qquad \forall \|\mathbf{W}_i\|_F \le 1, i \in [L-2], \|\mathbf{w}_{L-1}\|_2 \le 1.
\end{aligned}
\tag{91}
$$

### E.4 Proof of Theorem 6

The proof is analogous to the proof of Theorem 4. We can rewrite the dual problem as

$$
\begin{aligned}
D_{\text{ReLU}}^{\text{prl}} &= \max_{\mathbf{\Lambda}} \operatorname{tr}(\mathbf{\Lambda}^T \mathbf{Y}), \\
&\quad \text{s.t. } \|\mathbf{\Lambda}^T(((\mathbf{X}\mathbf{W}_1)_+ \ldots \mathbf{W}_{L-2})_+ \mathbf{w}_{L-1})_+\|_2 \le L/2, \\
&\qquad \forall (\mathbf{W}_1,\ldots,\mathbf{W}_{L-2},\mathbf{w}_{L-1}) \in \Theta,
\end{aligned}
\tag{92}
$$

where the set $\Theta$ is defined as

$$
\Theta = \{(\mathbf{W}_1,\ldots,\mathbf{W}_{L-2},\mathbf{w}_{L-1}) | \|\mathbf{W}_l\|_F \le 1, l \in [L-2], \|\mathbf{w}_{L-1}\|_2 \le 1\}.
\tag{93}
$$

By writing $\theta = (\mathbf{W}_1,\ldots,\mathbf{W}_{L-2},\mathbf{w}_{L-1})$, the bi-dual problem, i.e., the dual problem of (92), is given by

$$
\begin{aligned}
&\min \|\boldsymbol{\mu}\|_{\text{TV}}, \\
&\text{s.t. } \int_{\theta \in \Theta}(((\mathbf{X}\mathbf{W}_1)_+ \ldots \mathbf{W}_{L-2})_+ \mathbf{w}_{L-1})_+ d\boldsymbol{\mu}(\theta) = \mathbf{Y}.
\end{aligned}
\tag{94}
$$

Here $\boldsymbol{\mu}: \Sigma \to \mathbb{R}^K$ is a signed vector measure, where $\Sigma$ is a $\sigma$-field of subsets of $\Theta$ and $\|\boldsymbol{\mu}\|_{\text{TV}}$ is its total variation. The formulation in (94) has infinite width in each layer. According to Theorem 10 in Appendix G, the measure $\boldsymbol{\mu}$ in the integral can be represented by finitely many Dirac delta functions. Therefore, there exists $m^* \le KN + 1$ such that we can rewrite the problem (94) as

$$
\begin{aligned}
&\min \sum_{j=1}^{m^*} \|\mathbf{w}_{L,j}^{\text{row}}\|_2, \\
&\text{s.t. } \sum_{j=1}^{m^*}(((\mathbf{X}\mathbf{W}_{1,j})_+ \ldots \mathbf{W}_{L-2,j})_+ \mathbf{w}_{L-1,j}^{\text{col}})_+ \mathbf{w}_{L,j}^{\text{row}} = \mathbf{Y}, \\
&\quad \|\mathbf{W}_{l,j}\|_F \le 1, l \in [L-2], \|\mathbf{w}_{L-1,j}^{\text{col}}\|_2 \le 1, j \in [m^*].
\end{aligned}
\tag{95}
$$

Here the variables are $\mathbf{W}_{l,j}$ for $l \in [L-2]$ and $j \in [m^*]$, $\mathbf{W}_{L-1}$ and $\mathbf{W}_L$. This is equivalent to (23). As the strong duality holds for the problem (92) and (94), the primal problem (23) is equivalent to the dual problem (92) as long as $m \ge m^*$.

### E.5    PROOF OF THEOREM 7

Consider the following dual problem

$$D_{\text{ReLU}}^{\text{prl,sub}} = \max \ \text{tr}(\boldsymbol{\Lambda}^T \mathbf{Y}), \ \text{s.t.} \ \max_{i \in [m^*]} \|\boldsymbol{\Lambda}^T \mathbf{v}_i\|_2 \le L/2. \tag{96}$$

Apparently we have $D_{\text{ReLU}}^{\text{prl,sub}} \le D_{\text{ReLU}}^{\text{prl}}$. As $\boldsymbol{\Lambda}^*$ is the optimal solution to $D_{\text{ReLU}}^{\text{prl}}$ and $\boldsymbol{\Lambda}^*$ is feasible to $D_{\text{ReLU}}^{\text{prl,sub}}$, we have $D_{\text{ReLU}}^{\text{prl,sub}} \ge D_{\text{ReLU}}^{\text{prl}}$. This implies that $D_{\text{ReLU}}^{\text{prl,sub}} = D_{\text{ReLU}}^{\text{prl}}$. We note that (26) is the dual problem of (96). Therefore, as a corollary of Theorem 6, we have

$$P_{\text{ReLU}}^{\text{prl,sub}} = D_{\text{ReLU}}^{\text{prl,sub}} = D_{\text{ReLU}}^{\text{prl}} = P_{\text{ReLU}}^{\text{prl}}.$$

Therefore, $(\mathbf{W}_1, \dots, \mathbf{W}_L)$ is the optimal solution to (23).

## F    PROOFS OF AUXILIARY RESULTS

### F.1    PROOF OF LEMMA 1

Denote $a \in \mathbb{R}^n$ such that $a_i = A_{ii}$ and denote $b \in \mathbb{R}^n$ such that $b_i = \lambda_i(A)$. We can show that $a$ is majorized by $b$, i.e., for $k \in [n-1]$, we have

$$\sum_{i=1}^{k} a_{(i)} \le \sum_{i=1}^{k} b_{(i)}, \tag{97}$$

and $\sum_{i=1}^{n} a_i = \sum_{i=1}^{n} b_i$. Here $a_{(i)}$ is the $i$-th largest entry in $a$. We first note that

$$\sum_{i=1}^{n} A_{ii} = \text{tr}(A) = \sum_{i=1}^{n} \lambda_i(A).$$

On the other hand, for $k \in [n-1]$, we have

$$\begin{aligned}
\sum_{i=1}^{k} a_{(i)} &= \max_{v \in \mathbb{R}^n, v_i \in \{0,1\}, 1^T v = k} v^T a \\
&= \max_{v \in \mathbb{R}^n, v_i \in \{0,1\}, 1^T v = k} \text{tr}(\mathbf{diag}(v) A \mathbf{diag}(v)) \\
&\le \max_{V \in \mathbb{R}^{k \times n}, VV^T = I} \text{tr}(V A V^T) \\
&= \sum_{i=1}^{k} \lambda_i(A) = \sum_{i=1}^{k} b_{(i)}.
\end{aligned} \tag{98}$$

Therefore, $a$ is majorized by $b$. As $f(x) = -x^p$ is a convex function, according to the Karamata's inequality, we have

$$\sum_{i=1}^{n} f(a_i) \le \sum_{i=1}^{n} f(b_i).$$

This completes the proof.

### F.2    PROOF OF LEMMA 2

According to the min-max principle for singular value, we have

$$\sigma_i(W) = \min_{\dim(S) = d-i+1} \max_{x \in S, \|x\|_2 = 1} \|Wx\|_2.$$

As $P$ is a projection matrix, for arbitrary $x \in \mathbb{R}^d$, we have $\|PWx\|_2 \le \|Wx\|_2$. Therefore, we have

$$\max_{x \in S, \|x\|_2 = 1} \|PWx\|_2 \le \max_{x \in S, \|x\|_2 = 1} \|Wx\|_2.$$

This completes the proof.

## G    CARATHEODORY'S THEOREM AND FINITE REPRESENTATION

We first review a generalized version of Caratheodory's theorem introduced in (Rosset et al., 2007).

**Theorem 8** *Let $\mu$ be a positive measure supported on a bounded subset $D \subseteq \mathbb{R}^N$. Then, there exists a measure $\nu$ whose support is a finite subset of $D$, $\{z_1, \ldots, z_k\}$, with $k \leq N + 1$ such that*

$$\int_D z d\mu(z) = \sum_{i=1}^k z_i d\nu(z_i), \tag{99}$$

*and $\|\mu\|_{TV} = \|\nu\|_{TV}$.*

We can generalize this theorem to signed vector measures.

**Theorem 9** *Let $\mu : \Sigma \to \mathbb{R}^K$ be a signed vector measure supported on a bounded subset $D \subseteq \mathbb{R}^N$. Here $\Sigma$ is a $\sigma$-field of subsets of $D$. Then, there exists a measure $\nu$ whose support is a finite subset of $D$, $\{z_1, \ldots, z_k\}$, with $k \leq KN + 1$ such that*

$$\int_D z d\mu(z) = \sum_{i=1}^k z_i d\nu(z_i), \tag{100}$$

*and $\|\nu\|_{TV} = \|\mu\|_{TV}$.*

PROOF  Let $\mu$ be a signed vector measure supported on a bounded subset $D \subseteq \mathbb{R}^N$. Consider the extended set $\tilde{D} = \{zu^T | z \in D, u \in \mathbb{R}^K, \|u\|_2 = 1\}$. Then, $\mu$ corresponds to a scalar-valued measure $\tilde{\mu}$ on the set $\tilde{D}$ and $\|\mu\|_{\text{TV}} = \|\tilde{\mu}\|_{\text{TV}}$. We note that $\tilde{D}$ is also bounded. Therefore, by applying Theorem 8 to the set $\tilde{D}$ and the measure $\tilde{\mu}$, there exists a measure $\tilde{\nu}$ whose support is a finite subset of $\tilde{D}$, $\{z_1 u_1^T, \ldots, z_k u_k^T\}$, with $k \leq KN + 1$ such that

$$\int_{\tilde{D}} Z d\tilde{\mu}(Z) = \sum_{i=1}^k z_i u_i^T d\tilde{\nu}(z_i u_i^T), \tag{101}$$

and $\|\tilde{\mu}\|_{\text{TV}} = \|\tilde{\nu}\|_{\text{TV}}$. We can define $\nu$ as the signed vector measure whose support is a finite subset $\{z_1, \ldots, z_k\}$ and $d\nu(z_i) = u_i d\tilde{\nu}(z_i u_i)$. Then, $\|\nu\|_{\text{TV}} = \|\tilde{\nu}\|_{\text{TV}} = \|\tilde{\mu}\|_{\text{TV}} = \|\mu\|_{\text{TV}}$. This completes the proof.

Now we are ready to present the theorem about the finite representation of a signed-vector measure.

**Theorem 10** *Suppose that $\theta$ is the parameter with a bounded domain $\Theta \subseteq \mathbb{R}^p$ and $\phi(\mathbf{X}, \theta) : \mathbb{R}^{N \times d} \times \Theta \to \mathbb{R}^N$ is an embedding of the parameter into the feature space. Consider the following optimization problem*

$$\min \|\boldsymbol{\mu}\|_{TV}, \ s.t. \ \int_\Theta \phi(X, \theta) d\boldsymbol{\mu}(\theta) = Y. \tag{102}$$

*Assume that an optimal solution to (102) exists. Then, there exists an optimal solution $\hat{\boldsymbol{\mu}}$ supported on at most $KN + 1$ features in $\Theta$.*

PROOF  Let $\hat{\boldsymbol{\mu}}$ be an optimal solution to (102). We can define a measure $\hat{P}$ on $\mathbb{R}^N$ as the push-forward of $\hat{\boldsymbol{\mu}}$ by $\hat{P}(B) = \hat{\boldsymbol{\mu}}(\{\theta | \phi(X, \theta) \in B\})$. Denote $D = \{\phi(X, \theta) | \theta \in \Theta\}$. We note that $\hat{P}$ is supported on $D$ and $D$ is bounded. By applying Theorem 9 to the set $D$ and the measure $\hat{P}$, we can find a measure $Q$ whose support is a finite subset of $D$, $\{z_1, \ldots, z_k\}$ with $k \leq KN + 1$. For each $z_i \in D$, we can find $\theta_i$ such that $\phi(X, \theta_i) = z_i$. Then, $\tilde{\boldsymbol{\mu}} = \sum_{i=1}^k \delta(\theta - \theta_i) dQ(z_i)$ is an optimal solution to (102) with at most $KN + 1$ features and $\|\tilde{\boldsymbol{\mu}}\|_{\text{TV}} = \|\boldsymbol{\mu}\|_{\text{TV}}$. Here $\delta(\cdot)$ is the Dirac delta measure.

