# OpenReview forum: "Parallel Deep Neural Networks Have Zero Duality Gap"
_ICLR.cc/2023/Conference — ICLR 2023 poster_

### Official Review · Reviewer_BG6i · 2022-10-24

**Confidence:** 3
**Correctness:** 3
**Technical Novelty And Significance:** 3
**Empirical Novelty And Significance:** Not applicable
**Recommendation:** 6

**Clarity, Quality, Novelty And Reproducibility:**

This paper is an improvement to the works by Ergen & Pilanci 2020, 2021a and proves novel results on the duality of neural networks. The paper is mostly well-writen and easy to follow.

**Strength And Weaknesses:**

This paper extended the results by Ergen 2020 and studied the dual problem and bi-dual problem of deep neural networks. The author proved that the bi-dual problem of deep neural networks are in the form of parallel neural networks, and proved that strong duality may not hold for standard deep neural networks. The author proposed a parallel nerual network which strong duality holds.
However, as for the parallel neural networks, the author used $\\|\cdot\\|\_F^L$ regularization, which is different from standard neural networks. It is not possible to tell whether strong duality comes from the parallel architecture or the difference in regularization. Could the author proved more discussion on this point?
Besides, proposition 3 may be flawed. The proof includes the term $\sum_{j=1}^m\\|W\_{L,j}\\|\_F^L$ while the statement includes $\\|W\_{L}\\|\_F^L=(\sum_{j=1}^m \\|W\_{L,j}\\|\_F^2)^{L/2}$ . These two terms are not equal unless $L=2$.

**Summary Of The Paper:**

This paper studies the duality gap of deep neural networks and parallel neural networks. Two kinds of activation functions are considered: identity function and ReLU function. The author reformulated the optimization problem to a convex problem using AM-GM inequality, and study their dual problem. The bi-dual problem of neural networks are parallel neural networks. The author proved that the duality gap of deep neural networks may not be zero except for some special cases, while the duality gap of parallel neural networks are zero.

**Summary Of The Review:**

This paper extended the results by Ergen & Pilanci 2020 and studied the duality gap of neural networks and parallel neural networks. However, the regularizer used in the two neural networks are different, which makes the statement that parallel neural networks have zero duality gap problematic.

---

### Official Review · Reviewer_RuMQ · 2022-10-26

**Confidence:** 3
**Correctness:** 3
**Technical Novelty And Significance:** 3
**Empirical Novelty And Significance:** Not applicable
**Recommendation:** 8

**Clarity, Quality, Novelty And Reproducibility:**

Quality:

The theoretical results are of high quality.

Clarity:

The paper is mostly well-written.

Novelty:

The paper contains several novel theoretical results.


**Strength And Weaknesses:**

Strengths:

1. Solid theoretical results.
2. Novel results for convex formulations of neural networks.
3. The paper is well written.

Weaknesses:

1. There are no experiments on the performance of parallel networks and whether the convex optimization problems can be solved in practice.
2. It is not clear if the results on parallel linear networks in Section 3.2 are new. Table 1 shows that they were studied in previous works.
3. The process of obtaining the solution to the primal problem from the solution of the dual problem is not explained. Can the authors elaborate on this?


**Summary Of The Paper:**

This work shows new results on convex formulations of neural networks. The goal is to formulate the nonconvex optimization problem as minimum norm problems, derive the convex dual problem and study whether the duality gap is zero. If it is zero, then the dual can be solved via convex optimization tools with theoretical guarantees, instead of the nonconvex problem. For linear networks, it is shown that standard networks of depth at least 3 have non-zero duality gap. In contrast parallel linear architectures have zero duality gap. A similar result holds for networks with ReLU activations.

**Summary Of The Review:**

This paper shows novel theoretical results on convex formulations of neural networks. Experiments would improve the paper and clarify its practical significance.

---

### Official Review · Reviewer_jwcy · 2022-10-27

**Confidence:** 3
**Correctness:** 4
**Technical Novelty And Significance:** 3
**Empirical Novelty And Significance:** Not applicable
**Recommendation:** 6

**Clarity, Quality, Novelty And Reproducibility:**

#### Clarity
- When the text usually says: "the duality gap is non-zero", the authors should make clear they mean the duality gap is not guaranteed to be zero.
- At the end of Sec. 1.4., it says "Provided that $m \ge m^*$, where $m^* \le N + 1$", the introduction of $m^*$ is very confusing, as it is imposing $m$ to be greater than a lower bound, which seems meaningless ($m^* \le 0$). Some context is missing. This is much clearly stated in Theorem 4, when it says "There exists a critical width $m^*$...".
- A key assumption is that $m_l \ge \max (d, K), \forall l \in [L − 2]$. More insight on this wide enough constraint is missing. Is this a light version of assuming an overparameterised network?
- In (10), it is not clear the meaning of "aligned weights". What does it mean "the weight matrices in each branch"? The constraint shows the norm has the same upper bound $t$ independent on the branch or the layer (with the exception of $L$).
- Proposition 2 uses $\Sigma$ without introduction. Is this a diagonal matrix? Is this nonnegative? Is $W = U\Sigma V^\top$ the singular value decomposition? Please introduce it properly to avoid confusion.
- Similarly, Theorem 2 uses notation $X^\dagger$, but it is not until Theorem 3 when this is introduced as the pseudo-inverse. Please introduce the notation in Theorem 2.
- What does it mean "the bi-dual problem defined in (25) indeed optimizes with a parallel neural network"?
- What do you mean by "general regularised" in "We believe that our results can be easily generalized to general regularized training problems."? do you mean any regularisation?

Typos:
- "In Figure 1 and 2"... Figures
- At the end of page 6, ", two formulations (16) and (16)"

#### Quality
One main strategy roughly consists in formulating the dual problems, computing the respective optimal dual value in closed form, which is then compared with that of the primal problem. However, the authors don't justify the existence of optimal dual solutions. To be rigorous, the authors should prove that some regularity conditions hold that guarantee existence of optimal dual variables.

Apart from that, the paper seems sound, though I haven't checked the demonstrations in detail.

#### Novelty
The results are relevant but seem incremental. It is not clear whether any major technical contribution was needed to derive them.

#### Reproducibility
The authors provide proofs of all its results. Please make an effort to avoid skipping steps, like the following:
- For completeness when using Theorem 3 to state the primal and dual optimal are different, please add a demonstration (or ref) that they can only be equal when all singular values are equal.
- In the Appendix, below (58), please add a reference (or demonstration) that proves the inequality between the nuclear and Shatten-p norms.

**Strength And Weaknesses:**

#### Strengths
- The paper tackles the challenge of understanding the optimisation problem of training neural networks by focusing on the duality gap. This is an important area of research, with the high potential impact of making training deep learning models more easily and efficiently.
- The results are consistent with previous works and extend the state of the art.

#### Weaknesses
- The results seem incremental with respect to previous works.
- The authors don't make clear why previous works couldn't analyse the current case, and what the key novel technical contributions are that allow the current results, so it is difficult to evaluate the contribution.
- As a minor comment, there are some clarity issues that should be easily addressed, see below.

**Summary Of The Paper:**

This is a theoretical paper that extends previous results on the duality gap of neural network training. Its main results are: 1) linear networks with L2 regularization and 3 or more layers have in general positive duality gap, though its primal and dual solution can be computed in closed form; and 2) strong duality holds in parallel ReLU networks with a specific regularization of more than 3 layers.

**Summary Of The Review:**

This paper tackles an important problem and generalises previous results. Although the extension is incremental, it is general in the number of layers and seems to be a steppingstone towards more general results. On the other hand, it is not clear whether the authors have to innovate in the proof techniques.

All in all, the paper can be interesting for a part of the community studying this particular problem, and though it might not have an immediate impact in the community at large, it will hopefully inspire future research to study the practical impact of this research (by, e.g., leveraging fast parallel convex optimisers for practical applications of deep parallel networks.)

---

### Official Review · Reviewer_Lv6H · 2022-11-04

**Confidence:** 4
**Correctness:** 3
**Technical Novelty And Significance:** 2
**Empirical Novelty And Significance:** 1
**Recommendation:** 3

**Clarity, Quality, Novelty And Reproducibility:**

Some mathematical details are not rigorously treated:
- "there exists a feasible solution for such that $\phi(XW_1)W_2 = Y$": do you mean a feasible point? Or a minimizer? If you mean that some minimizers, perfectly fit the data, this needs to be shown, as it is not trivial. For example, for overparametrized Ridge Regression, this does not hold. It does not seem true to me that 3 is equivalent to 4, yet the paper studies 4 as a proxy 3. Can the authors comment ?


I found the paper to be very hard to read. In particular:

- Several claims are highly debatable:
    - "This [non zero duality gap] imposes more difficulty to train deep neural networks": the proposed convex reformulation do not lead to practical algorithm, do they ? If so, can the authors propose an experiment on real data with a practical method? Otherwise, it is unclear how even a zero duality gap helps.
    - the dual problem is a convex optimization problem the can be solved efficiently: this is by far not always true, some convex problems are hard to solve.

- The introduction cites a high number of paper without really explaining why they are relevant to the literature review, e.g.:
    - "The implicit regularization in training deep linear networks": what does this have to do with the rest of the introduction?

- I found the paper to be hard to read due to numerous grammar mistakes. I suggest the author carefully proofread their paper.
    - For convex optimization problems, the convex duality is an important
    tool to determine its optimal value: what does "its" refer to here ? "problems" is plural

    - the authors need to distinguish between \citet (when the citation is part of the sentence) and \citep (when it is not). "In (Paternain et al., 2019)," should use citet, not citep
    - the zero duality gap is hard to achiev: **a** zero
    - the authors need to be careful to distinguish between "a" and "the", e.g. in "neural networks with the parallel architecture" it should be "a", or nothing, but not "the".
    - Same in "The parallel models with overparameterization are", there should be "The".
    - Same in "the strong duality holds", "We then introduce the neural network with the parallel architecture", ...
    - Shatten > Schatten


**Strength And Weaknesses:**

Strength:
- the convex reformulation of neural networks with path or L2 regularization is an interesting direction of research that has attracted attention lately, from the theoretical point of view.

Weaknesses:
- half of the paper is devoted to linear networks, that are not, to my knowledge, used in practice
- there is a stream of papers on the same topic with marginal improvements from one to the other (i.e., Ergen and Pilanci 2021e) and the contribution here may seem incremental. The proofs techniques are based on already known dual reformulations by Ergen and Pilanci; the main novelty to me is Proposition 2 that is easy to prove.
- the proposed approach is not practical (for example, considering rank 1 design matrices); there are not experiments in the paper.

**Summary Of The Paper:**

It has been recently been shown that 2 layers L2 regularized relu networks are equivalent to convex problems and have "zero duality gap".
The paper studies a generalization of this result: does it hold when the network's output is vectorial, and for more than 3 layers?
The paper shows that some regularized neural networks with more than 2 layers and vector output have a non zero duality gap.
The authors prove that this duality gap vanishes for so-called "parallel architectures", with a particular regularization.

**Summary Of The Review:**

Marginal improvement over literature on the theoretical side, no practical contribution on the the practical one.

---

### Decision · Program_Chairs · 2023-01-20

**Decision:**

Accept: poster

**Justification For Why Not Higher Score:**

See the discussion above.

**Justification For Why Not Lower Score:**

I think the paper meets the ICLR bar.

**Metareview: Summary, Strengths And Weaknesses:**

The paper proves three main results

1.  The standard deep **linear** neural networks does not satisfy strong duality in general beyond two layers.

2. Parallel neural networks with **linear activations** and a specific convex regularization term enjoys strong duality **in the interpolation regime**, thus (by taking the dual of the dual) can be converted into a convex program.

3. A deep parallel **ReLU-activated** neural network with the same convex regularization has zero duality gap **in the interpolation regime**.

While the majority of the reviewers find the results interesting, there are also a few important weakness of the manuscript reviewers have highlighted.

* Linear DNNs are not used and are of much less theoretical interests
* Results on DNNs with Rank 1 covariate matrix are restrictive.
* The results require interpolation and do not cover non-interpolating cases.
* The technical novelty is low (while the results are new, the proof techniques are mostly from existing work)
* Some statements are not carefully qualified, e.g., "convex thus efficiency solvable".



**Note From Pc:**

if the above contains the word "oral" or "spotlight" please see: "oral" presentation means -> notable-top-5% and "spotlight" means -> notable-top-25%. As stated in our emails, we are disassociating presentation type from AC recommendations

**Summary Of Ac-Reviewer Meeting:**

Both the strength and weaknesses of the work are prominent. The reviewers find the research direction promising and the results interesting despite being slightly incremental. Overall, I find the paper a worthy contribution and would happily support accepting this paper to ICLR.

The final version should resolve the issues raised by the reviewers and to discuss the results more accurately in the paper.
Table 1 is a bit confusing. I suggest replacing the crosses with question marks to indicate that the problems are still open, highlighting the contribution of the current paper in a different color, and adding footnotes to indicate the additional assumptions needed for strong duality to be true  (e.g., whether interpolation is needed).